# ATM and CDK2 control chromatin remodeler CSB to inhibit RIF1 in DSB repair pathway choice

Nicole L. Batenburg [1], John R. Walker [1], Sylvie M. Noordermeer [2,3], Nathalie Moatti[2], Daniel Durocher [2] & Xu-Dong Zhu[1]

CSB, a member of the SWI2/SNF2 superfamily, is implicated in DNA double-strand break (DSB) repair. However, how it regulates this repair process is poorly understood. Here we uncover that CSB interacts via its newly identified winged helix domain with RIF1, an effector of 53BP1, and that this interaction mediates CSB recruitment to DSBs in S phase. At DSBs, CSB remodels chromatin by evicting histones, which limits RIF1 and its effector MAD2L2 but promotes BRCA1 accumulation. The chromatin remodeling activity of CSB requires not only damage-induced phosphorylation on S10 by ATM but also cell cycle-dependent phosphorylation on S158 by cyclin A-CDK2. Both modifications modulate the interaction of the CSB N-terminal region with its ATPase domain, the activity of which has been previously reported to be autorepressed by the N-terminal region. These results suggest that ATM and CDK2 control the chromatin remodeling activity of CSB in the regulation of DSB repair pathway choice.

---

[1] Department of Biology, McMaster University, Hamilton, Ontario L8S 4K1, Canada. [2] Lunenfeld-Tanenbaum Research Institute, Mount Sinai Hospital, 600 University Avenue, Toronto, Ontario M5G 1X5, Canada. [3] Present address: Department of Human Genetics, Leiden University Medical Centre, Einthovenweg 20, 2333 ZC Leiden, The Netherlands. Correspondence and requests for materials should be addressed to X.-D.Z. (email: zhuxu@mcmaster.ca)

DNA double-strand breaks (DSBs), one of the most lethal forms of DNA damage, can threaten genomic integrity and promote tumorigenesis or premature aging if not repaired properly. Eukaryotic cells have evolved two mechanistically distinct pathways to repair DSBs: nonhomologous end joining (NHEJ) and homologous recombination (HR)[1, 2]. NHEJ can ligate two broken ends in the absence of sequence homology whereas HR uses homologous sequences as a template to repair broken DNA. While NHEJ is active throughout interphase, HR is primarily confined to S and G2 phases when sister chromatids are present. The choice of DSB repair pathways is highly regulated during the cell cycle, with two proteins 53BP1 and BRCA1 playing pivotal but antagonzing roles in this process[3–7]. 53BP1 blocks BRCA1 and promotes NHEJ in G1 through its downstream effector RIF1[8–12]. Phosphorylation of 53BP1 by ATM on its N-terminal region promotes RIF1 recruitment to DSBs, which prevents DNA end resection and channels DSBs towards NHEJ. In S/G2 phases, BRCA1 antagonizes 53BP1, perhaps through repositioning 53BP1 on the damaged chromatin[3, 13]. BRCA1 also blocks RIF1 from DSBs in S phase[8–10, 14], paving the way for the initiation of DNA end resection.

**Fig. 1** CSB interacts with RIF1 and is recruited by RIF1 to FokI-induced DSBs. **a** Immunofluorescence of U2OS-265 cells with or without induction of FokI expression. Fixed cells were stained with anti-CSB antibody. Nuclei were stained with DAPI in blue in this and following figures. Scale bars, 5 μm. **b** Immunofluorescence of U2OS-265 cells expressing Myc-tagged CSB with or without induction of FokI expression. Fixed cells were stained with anti-Myc antibody. Scale bars, 5 μm. **c** Quantification of the percentage of U2OS-265 cells with Myc-CSB accumulated at FokI-induced DSBs. U2OS-265 cells were treated with DMSO or ATM inhibitor KU55933 for 1 h prior to induction of FokI expression. A total of 250 Myc-expressing cells were scored for each independent experiment in a blind manner. Standard deviations, referred to as SDs in this and the subsequent figures, from three independent experiments are indicated. *$P < 0.05$ (Student $t$ test). **d** Quantification of the percentage of parental and 53BP1 KO U2OS-265 cells with Myc-CSB accumulated at FokI-induced DSBs. Scoring was done as described in 1c. SDs from three independent experiments are indicated. *$P < 0.05$ (Student $t$ test). **e** Immunofluorescence with an anti-Myc antibody. 48 h prior to FokI induction, U2OS-265 cells were transfected with siRNA against scramble DNA (siControl) or RIF1 (siRIF1). Scale bars, 5 μm. **f** Quantification of the percentage of siControl- and siRIF1-expressing U2OS-265 cells with Myc-CSB accumulated at FokI-induced DSBs. Scoring was done as described in 1c. SDs from three independent experiments are indicated. *$P < 0.05$ (Student $t$ test). **g** Quantification of the percentage of siControl- and siCSA-expressing U2OS-265 cells with Myc-CSB accumulated at FokI-induced DSBs. Scoring was done as described in 1c. SDs from three independent experiments are indicated. **h** Coimmunoprecipitation with IgG and anti-CSB antibody in HCT116 cells treated with or without 20 Gy IR. Immunoblotting was performed with anti-CSB, anti-RIF1 and anti-53BP1 antibodies. Protein molecular weight markers in kDa are indicated in this and the subsequent figures. **i** Coimmunoprecipitation with anti-RIF1 antibody in parental (WT) and CSB knockout (KO) HCT116 cells treated with or without 20 Gy IR. Immunoblotting was performed with anti-RIF1, anti-53BP1 and anti-CSB antibodies

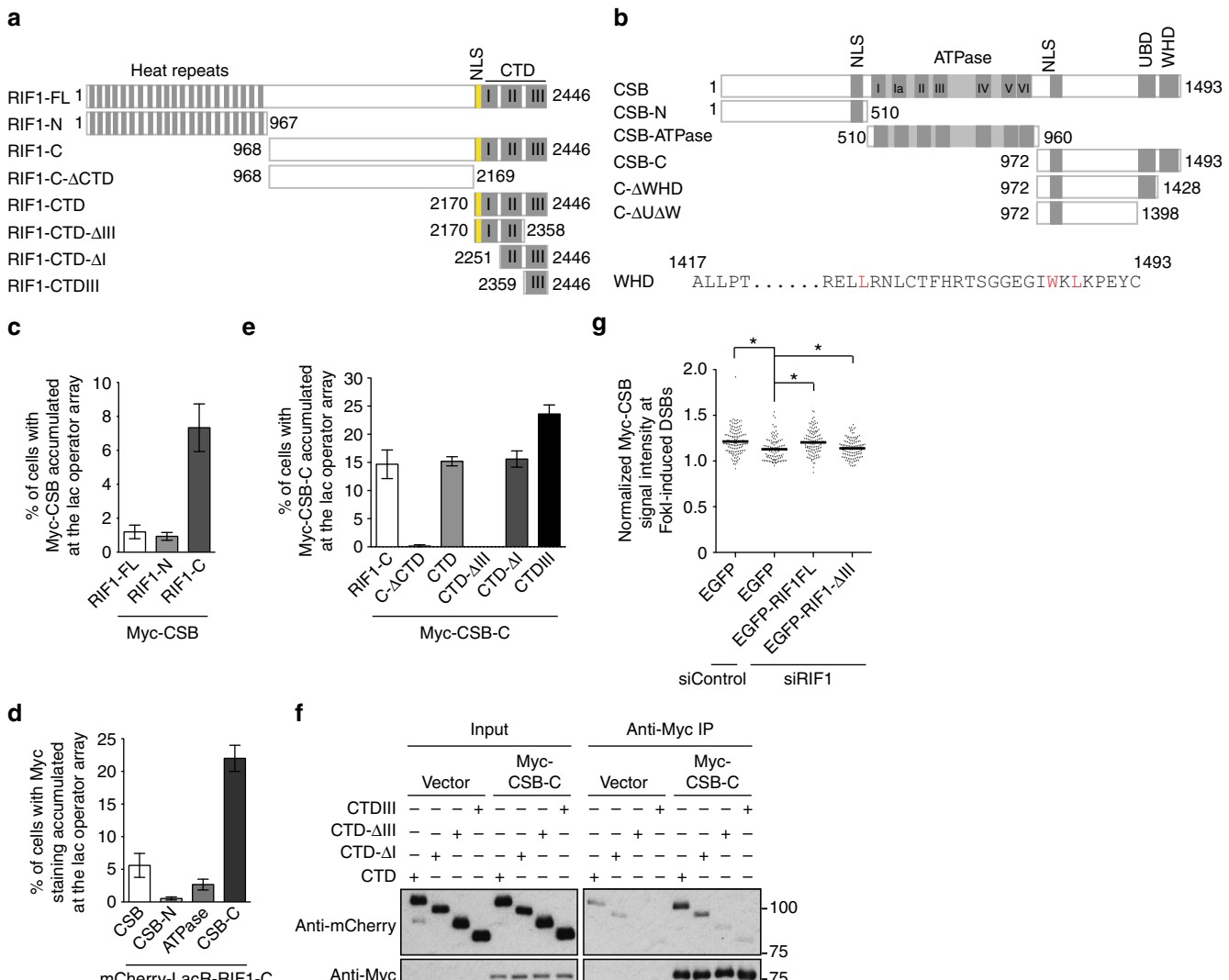

**Fig. 2** RIF1 interacts with CSB and recruits CSB to DSBs via its CTD domain. **a** Schematic diagram of RIF1. NLS: nuclear localization signal; CTD: C-terminal domain. **b** Schematic diagram of CSB. **c** Quantification of the percentage of cells exhibiting Myc-CSB accumulated at the lac operator array. U2OS-265 cells were co-transfected with Myc-CSB and various mCherry-LacR-RIF1 alleles as indicated. A total of 250 cells positive for Myc-CSB expression were scored for each independent experiment in a blind manner. SDs from three independent experiments are indicated. **d** Quantification of the percentage of cells exhibiting Myc staining accumulated at the lac operator array. U2OS-265 cells were co-transfected with mCherry-LacR-RIF1-C and various Myc-tagged CSB alleles as indicated. A total of 250 cells positive for expression of various Myc-tagged CSB alleles as indicated were scored for each independent experiment in a blind manner. SDs from three independent experiments are indicated. **e** Quantification of the percentage of cells exhibiting Myc-CSB-C accumulated at the lac operator array. U2OS-265 cells were co-transfected with Myc-CSB-C and various mCherry-LacR-RIF1-C alleles as indicated. A total of 250 cells positive for Myc-CSB-C expression were scored for each independent experiment in a blind manner. SDs from three independent experiments are indicated. **f** Coimmunoprecipitation with anti-Myc antibody in 293T cells expressing the vector alone or Myc-CSB-C in conjunction with various mCherry-LacR-CTD alleles as indicated. Immunoblotting was done with anti-Myc and anti-mCherry antibodies. **g** Quantification of the intensity of Myc-CSB signal at the site of FokI-induced DSBs. 24 h post transfection with siControl or siRIF1, U2OS-265 cells were transfected with the EGFP vector alone or various siRIF1-resistant EGFP-RIF1 alleles as indicated and induced for FokI expression 48 h post transfection. Analysis of Myc-CSB signal intensity was only done for cells positive for expression of Myc-CSB, EGFP and mCherry-LacR-FokI. The respective numbers of cells analyzed for siControl/EGFP, siRIF1/EGFP, siRIF1/EGFP-RIF1 and siRIF1/EGFP-Rif-DCTDIII were 131, 107, 120 and 124. *$P < 0.05$ (Mann–Whitney test). NLS nuclear localization signal; UBD ubiquitin-binding domain; WHD winged helix domain

Upon induction of DSBs, the chromatin structure needs to be modified to facilitate efficient access of repair factors to DSBs[15]. In mammalian cells, limited or local nucleosome disassembly occurs in G1 phase when DSBs are repaired by NHEJ whereas extensive nucleosome disassembly is associated with HR in S/G2 cells[16–19]. How nucleosome disassembly is controlled in a cell-cycle-dependent manner remains unclear. Many ATP-dependent chromatin remodeling complexes participate in chromatin disassembly to allow for efficient DSB repair[15]; however, the exact mechanism by which these complexes are regulated locally to remodel chromatin and to facilitate DSB repair remains poorly understood.

Cockayne syndrome (CS), a devastating hereditary disorder, is characterized by physical impairment, neurological degeneration and segmental premature aging. The majority of CS patients carry mutations in the *ERCC6* gene encoding Cockayne syndrome group B protein (CSB). CSB, a multifunctional protein, participates in a number of cellular processes, including transcription[20],

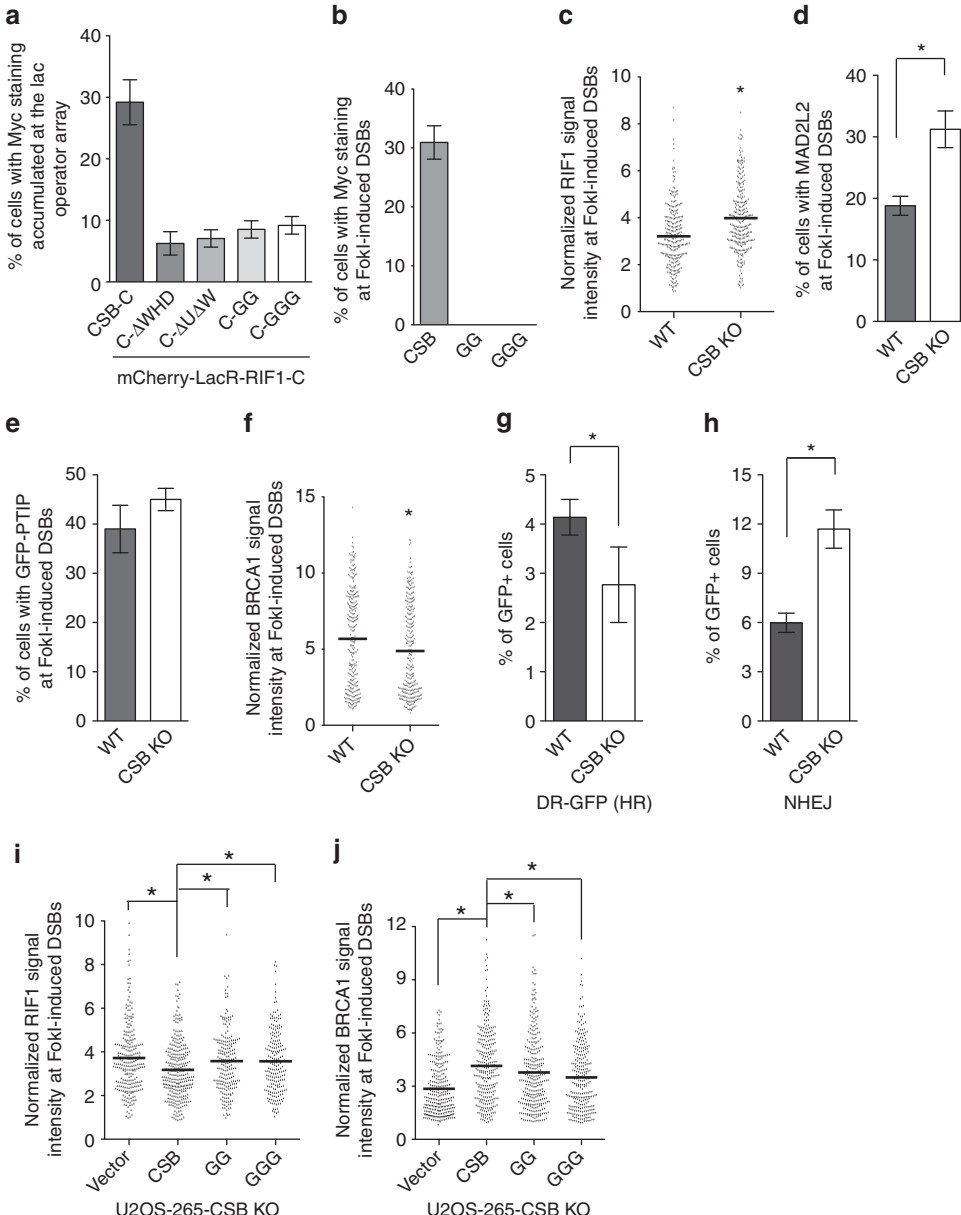

**Fig. 3** CSB interacts with RIF1 and inhibits RIF1 at DSBs. **a** Quantification of the percentage of cells exhibiting Myc staining accumulated at the lac operator array in U2OS-265 cells co-transfected with indicated alleles. A total of 250 cells positive for Myc staining were scored for each independent experiment in a blind manner. SDs from three independent experiments are indicated. **b** Quantification of the percentage of cells exhibiting Myc staining at the site of FokI-induced DSBs. A total of 250 U2OS-265 cells expressing various Myc-tagged CSB alleles as indicated were scored for each independent experiment in a blind manner. SDs from three independent experiments are indicated. **c** Quantification of the intensity of RIF1 signal at FokI-induced DSBs. The respective numbers of cells analyzed for parental and CSB KO were 275 and 277. *$P < 0.05$ (Mann–Whitney test). **d** Quantification of the percentage of cells exhibiting MAD2L2 at FokI-induced DSBs. A total of 500–550 cells were scored for each independent experiment in a blind manner. SDs from three independent experiments are indicated. *$P < 0.05$ (Student $t$ test). **e** Quantification of the percentage of cells exhibiting GFP-PTIP at FokI-induced DSBs. A total of 500 cells were scored for each independent experiment in a blind manner. SDs from three independent experiments are indicated. **f** Quantification of the intensity of BRCA1 signal at FokI-induced DSBs. The respective numbers of cells analyzed for parental and CSB KO were 282 and 294. *$P < 0.05$ (Mann–Whitney test). **g** HR-mediated repair of *I-SceI*-induced DSBs in U2OS-DR-GFP WT and CSB-KO cells. SDs from three independent experiments are indicated. *$P < 0.05$ (Student $t$ test). **h** NHEJ-mediated repair of *I-SceI*-induced DSBs. SDs from three independent experiments are indicated. *$P < 0.05$ (Student $t$ test). **i** Quantification of the intensity of RIF1 signal at FokI-induced DSBs. The respective numbers of cells analyzed for the vector alone, Myc-CSB, Myc-CSB-GG and Myc-CSB-GGG were 298, 304, 204, and 209. *$P < 0.05$ (Mann–Whitney test). **j** Quantification of the intensity of BRCA1 signal at FokI-induced DSBs. The respective numbers of cells analyzed for the vector alone, Myc-CSB, Myc-CSB-GG and Myc-CSB-GGG were 279, 270, 291, 258. *$P < 0.05$ (Mann–Whitney test)

transcription-coupled repair[21, 22], oxidative damage[23], mitochondria function[24, 25], telomere maintenance[26] and DSB repair[27–29]. CSB forms IR-induced damage foci and regulates DSB repair pathway choice[27]. Loss of CSB induces RIF1 accumulation at DSBs specifically in S/G2 cells[27], thereby hindering BRCA1 recruitment to DSBs. However, how CSB is recruited to DSBs and what it does at DSBs to facilitate efficient HR remains unclear. CSB contains a central SWI2/SNF2-like ATPase domain and its in vitro ATPase activity is autoinhibited by its N-terminal region[30, 31], but the physiological mechanism that promotes its ATPase activity is unknown. Furthermore, CSB possesses ATP-dependent chromatin remodeling activity in vitro[30, 32, 33]; however, whether CSB may function as a chromatin remodeler in vivo has not yet been demonstrated.

Here we uncover that CSB interacts with RIF1 and is recruited by RIF1 to DSBs in S/G2. This interaction is modulated by the C-terminal domain (CTD) of RIF1 and a newly identified winged helix domain (WHD) at the C-terminus of CSB. We demonstrate that CSB is a chromatin remodeler in vivo, evicting histones from chromatin surrounding DSBs. The N-terminus of CSB is necessary for its chromatin remodeling activity, disruption of which induces RIF1 accumulation at DSBs in S/G2 but impairs BRCA1, RAD51 and HR. The chromatin remodeling activity of CSB at DSBs is controlled by two phosphorylation events, one being damage-induced S10 phosphorylation by ATM and the other being cell-cycle-regulated S158 phosphorylation by cyclin A-CDK2. Both S10 and S158 phosphorylations modulate the interaction of CSB N-terminus with its ATPase domain. These results led us to propose that CSB phosphorylations by ATM and CDK2 function as molecular signals to unlock its chromatin remodeling activity, perhaps by releasing the autoinhibition of its N-terminus. Subsequent nucleosome disassembly by CSB at DSBs inhibits RIF1 and paves the way for BRCA1-mediated HR.

## Results

**RIF1 interacts with CSB and recruits it to DSBs**. To investigate the mechanism by which CSB is recruited to DSBs, we employed a well-established reporter osteosarcoma cell line U2OS-265[34], which has the 256 copy lac operator array integrated into a single site on chromosome 1p3.6. Overexpression of the FokI nuclease domain fused to mCherry-LacR (mCherry-LacR-FokI) in the reporter cells resulted in a robust production of DSBs within the lac operator array. Both endogenous CSB and Myc-CSB were found to accumulate at FokI-induced DSBs (Fig. 1a, b). On the other hand, we did not detect any significant accumulation of endogenous CSA, a factor known to be involved in transcription-coupled repair (TCR)[35], at FokI-induced DSBs (N. L. Batenburg and X.D. Zhu, unpublished data).

Myc-CSB accumulation at FokI-induced DSBs was sensitive to ATM inhibition (Fig. 1c), loss of 53BP1 (Fig. 1d) and RIF1 (Fig. 1e, f), but not to depletion of CSA (Fig. 1g). These results prompted us to investigate if CSB might interact with RIF1 since RIF1 recruitment to DSBs is entirely dependent upon ATM and 53BP1[8, 9, 11]. Coimmunoprecipitation with anti-CSB antibody in HCT116 cells brought down RIF1 but not 53BP1 (Fig. 1h). The CSB interaction with RIF1 was also confirmed in a reverse immunoprecipitation with anti-RIF1 antibody (Fig. 1i). The discrepancy between the amount of CSB brought down by anti-RIF1 antibody and the amount of RIF1 brought down by anti-CSB antibody may imply that CSB might not interact with RIF1 in a 1:1 stoichiometry, however we cannot rule out the possibility that this discrepancy might be due to a difference in IP efficiency. As a control, coimmunoprecipitation with anti-RIF1 antibody brought down 53BP1 but not CSB in CSB knockout HCT116 cells (Fig. 1i), suggesting that CSB interaction with RIF1 is specific.

Furthermore, treatment with ionizing radiation did not significantly affect CSB interaction with RIF1 (Fig. 1h, i). These results reveal that CSB interacts with RIF1 independently of not only 53BP1 but also damage induction.

To gain further insights into CSB interaction with RIF1, we returned to the reporter U2OS-265 cell line, which allows for analysis of protein–protein interactions with a bait protein fused to mCherry-LacR. Full length RIF1 and RIF1 deletion alleles (RIF1-N and RIF1-C) (Fig. 2a) were fused to mCherry-LacR. Their ability to recruit Myc-CSB, Myc-CSB deletion alleles (CSB-N, CSB-ATPase and CSB-C) (Fig. 2b, top panel) to the lac operator array was examined in U2OS-265 cells. We observed a robust interaction between mCherry-LacR–RIF1-C and Myc-CSB-C (Fig. 2c, d and Supplementary Fig. 1a, b). The level of expression of mCherry-LacR-RIF1-FL was much lower than that of mCherry-LacR-RIF1-N and mCherry-LacR-RIF1-C (Supplementary Fig. 1c), which likely contributed to the poor interaction observed between mCherry-LacR-RIF1-FL and Myc-CSB.

Deletion analysis revealed that the CTD of RIF1 was necessary and sufficient for its interaction with Myc-CSB-C (Fig. 2e, f and Supplementary Fig. 2a). While deletion of CTDI subdomain did not affect mCherry-LacR–RIF1-CTD interaction with Myc-CSB-C at the lac operator array (Fig. 2e and Supplementary Fig. 2a), it moderately affected the ability of mCherry-LacR-RIF1-CTD to coimmunoprecipitate with Myc-CSB-C (Fig. 2f). Deletion of the CTDIII subdomain abrogated the ability of mCherry-LacR-RIF1-CTD not only to interact with Myc-CSB-C at the lac operator array but also to coimmunoprecipitate with Myc-CSB-C (Fig. 2e, f and Supplementary Fig. 2a), suggesting that the CTDIII subdomain is necessary for RIF1 interaction with CSB. mCherry-LacR-CTDIII was observed to interact with Myc-CSB-C at the lac operator array (Fig. 2e, Supplementary Fig. 2a) but failed to coimmunoprecipitate Myc-CSB-C (Fig. 2f), the latter suggesting that CTDIII alone may not be sufficient to mediate RIF1 interaction with CSB. The discrepancy in the observed CTDIII interaction with CSB-C may be in part due to the difference in experimental conditions.

To investigate the role of the CTDIII subdomain of RIF1 in recruiting CSB to DSBs, we knocked down RIF1 in U2OS-265 cells and complemented RIF1-depleted cells with either the vector alone, siRIF1-resistant RIF1-FL or siRIF1-resistant RIF1-ΔCTDIII. RIF1 knockdown significantly affected CSB accumulation at FokI-induced DSBs (Fig. 2g). While introduction of EGFP-RIF1-FL rescued CSB accumulation at FokI-induced DSBs, overexpression of EGFP-RIF1-ΔCTDIII failed to do so (Fig. 2g and Supplementary Fig. 2b). These results suggest that the CTDIII subdomain of RIF1 is necessary for recruiting CSB to DSBs.

**CSB interacts with RIF1 via a newly identified WHD**. Deletion of the last 65 amino acids of CSB drastically affected Myc-CSB-C interaction with mCherry-LacR-RIF1-C in U2OS-265 cells (Fig. 3a and Supplementary Fig. 2c). Further deletion of previously described UBD domain[36] did not lead to any further reduction in Myc-CSB-C interaction with mCherry-LacR–RIF1-C (Fig. 3a and Supplementary Fig. 2c), suggesting that the last 65 amino acids of CSB is necessary for its interaction with RIF1. Profile–profile alignment and fold-recognition using the program FFAS[37] revealed that the last 76 amino acids of CSB resembled a winged helix domain (WHD) (Supplementary Fig. 3a). Additional protein threading trials using PHYRE (http://www.sbg.bio.ic.ac.uk/) strengthened our original prediction and further revealed that sequences belonging to more distantly related CSB homologs such as the yeast Rad26 were also likely to fold into the WHD, suggesting that this domain is evolutionarily conserved.

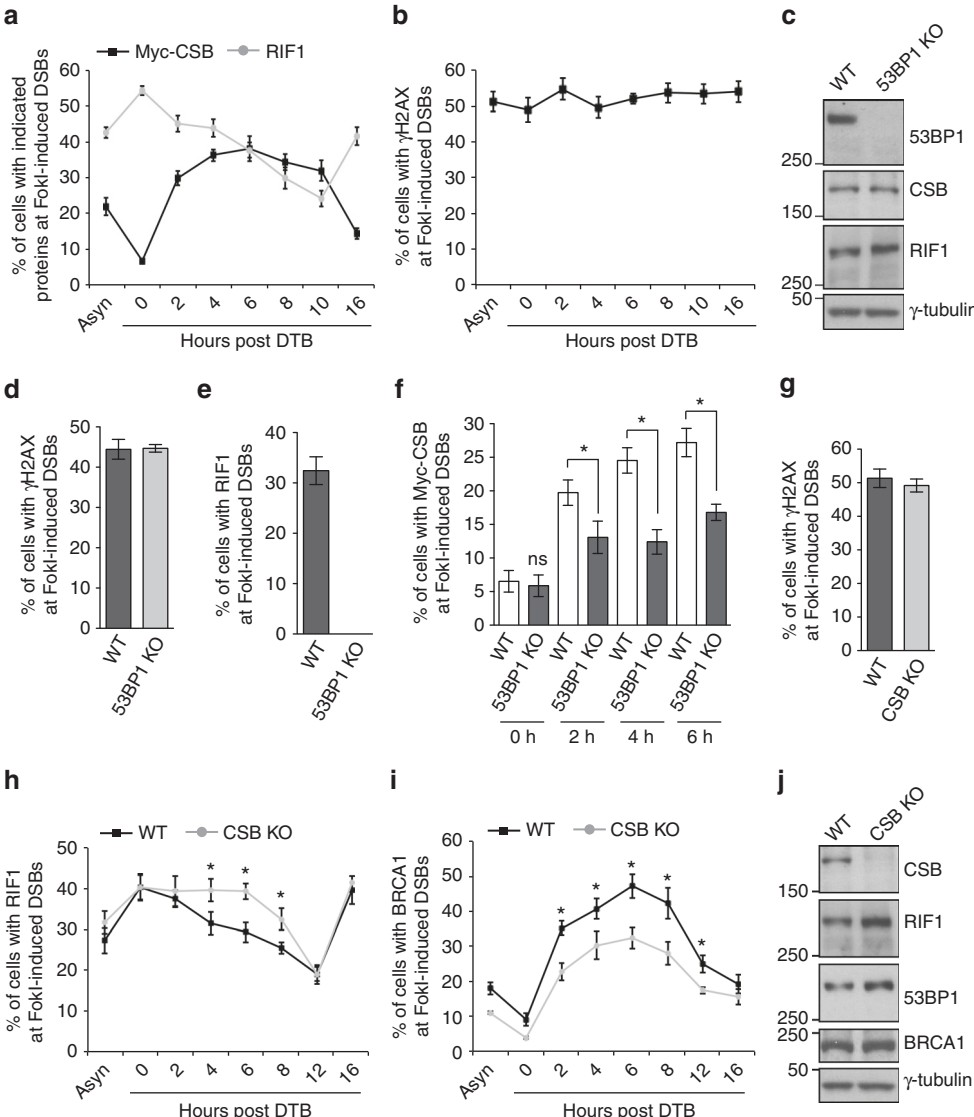

**Fig. 4** CSB is recruited to FokI-induced DSBs in S phase to inhibit RIF1. **a** Quantification of the percentage of synchronized Myc-CSB-expressing U2OS-265 cells exhibiting indicated proteins at FokI-induced DSBs. For Myc-CSB, a total of 250 Myc-CSB-expressing cells were scored for each independent experiment in blind. For RIF1, a total of 500–550 cells were scored in blind for each independent experiment. SDs from three independent experiments are indicated. **b** Quantification of the percentage of synchronized Myc-CSB-expressing U2OS-265 cells exhibiting γH2AX at FokI-induced DSBs. A total of 500–550 cells were scored in blind for each independent experiment. SDs from three independent experiments are indicated. **c** Western analysis of U2OS-265 parental (WT) and 53BP1 KO cells. The γ-tubulin blot was used as a loading control here and the following figures. **d** Quantification of the percentage of U2OS-265 WT and 53BP1 KO cells with γH2AX at FokI-induced DSBs. Scoring was done as in 3b. SDs from three independent experiments are indicated. **e** Quantification of the percentage of U2OS-265 WT and 53BP1 KO cells with RIF1 at FokI-induced DSBs. Scoring was done as in Fig 3b. SDs from three independent experiments are indicated. **f** Quantification of the percentage of synchronized Myc-CSB-expressing WT and 53BP1 KO U2OS-265 cells with Myc-CSB at FokI-induced DSBs. A total of 250 Myc-CSB-expressing cells were scored for each independent experiment in blind. SDs from three independent experiments are indicated. *$P < 0.05$ (Student $t$ test). **g** Quantification of the percentage of U2OS-265 WT and CSB-KO cells with γH2AX at FokI-induced DSBs. Scoring was done as in Fig. 3b. SDs from three independent experiments are indicated. **h** Quantification of the percentage of synchronized U2OS-265 WT and CSB KO cells with RIF1 at FokI-induced DSBs. Scoring was done as in Fig. 3b. SDs from three independent experiments are indicated. *$P < 0.05$ (Student $t$ test). **i** Quantification of the percentage of synchronized U2OS-265 WT and CSB KO cells with BRCA1 at FokI-induced DSBs. Scoring was done as in Fig. 3b. SDs from three independent experiments are indicated. *$P < 0.05$ (Student $t$ test). **j** Western analysis of U2OS-265 WT and CSB KO cells

Computer modeling of this domain on reported crystal structure of the WHD of the general transcription factor TFIIF[38] suggested that L1470, W1486 and L1488 of CSB, all of which are evolutionarily conserved (Supplementary Fig. 3b), contributed to the hydrophobic core formation of the CSB WHD (Supplementary Fig. 3c). To gain further insight into the role of this newly identified WHD, we generated CSB mutant alleles carrying simultaneous mutations of W1486 and L1488 to glycines (GG) or

simultaneous mutations of L1470, W1486, and L1488 to glycines (GGG). Both Myc-CSB-C-GG and Myc-CSB-C-GGG were severely defective in their interaction with mCherry-LacR-RIF1-C (Fig. 3a), indistinguishable from Myc-CSB-C lacking the WHD (Myc-CSB-C-ΔWHD) (Fig. 3a), suggesting that the WHD of CSB mediates its interaction with RIF1.

While Myc-CSB was readily recruited to FokI-induced DSBs, neither Myc-CSB-GG nor Myc-CSB-GGG were able to

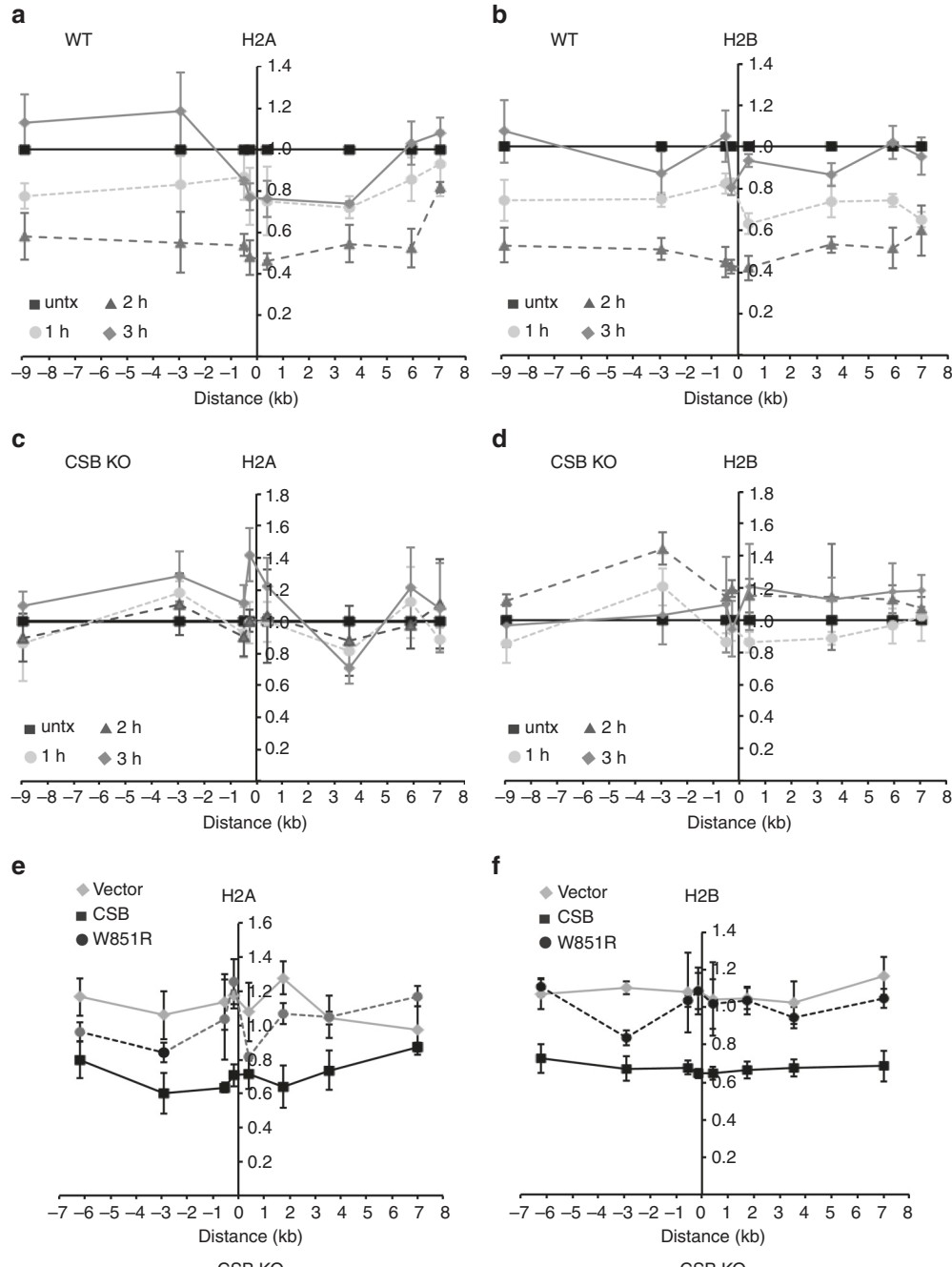

**Fig. 5** CSB evicts histones from the chromatin surrounding I-PpoI-induced DSBs in vivo. **a** Relative occupancy of histone H2A in ddI-PpoI-expressing hTERT-RPE parental cells. Cells were either untreated (untx) or treated with Shield-1 and 4-OHT and then collected at indicated times. The *x*-axis represents the distance in kb upstream and downstream from the I-PpoI-induced DSB on chromosome 1, which was set as 0. The *y*-axis represents the relative occupancy of H2A of treated cells relative to untreated cells. Standard error of the mean (SEM) from three independent experiments are indicated. **b** Relative occupancy of histone H2B in ddI-PpoI-expressing hTERT-RPE parental cells. Both *x*- and *y*-axes are as described in **a**. SEM from three independent experiments are indicated. **c** Relative occupancy of histone H2A in ddI-PpoI-expressing hTERT-RPE CSB KO cells. Both *x*- and *y*-axes are as described in **a**. SEM from three independent experiments are indicated. **d** Relative occupancy of histone H2B in ddI-PpoI-expressing hTERT-RPE CSB KO cells. Both *x*- and *y*-axes are as described in **a**. SEM from three independent experiments are indicated. **e** Relative occupancy of histone H2A in ddI-PpoI-expressing hTERT-RPE CSB KO cells complemented with the vector alone, Myc-tagged wild-type CSB or Myc-tagged mutant CSB-W851R. Both *x*- and *y*-axes are as described in 5a. SEM from three independent experiments are indicated. **f** Relative occupancy of histone H2B in ddI-PpoI-expressing hTERT-RPE CSB KO cells complemented with the vector alone, Myc-tagged wild-type CSB or Myc-tagged mutant CSB-W851R. Both *x*- and *y*-axes are as described in **a**. SEM from three independent experiments are indicated

accumulate at FokI-induced DSBs (Fig. 3b and Supplementary Fig. 2d), underscoring the importance of the WHD in mediating CSB accumulation at DSBs.

In agreement with the previous report that CSB regulates DSB pathway choice[27], knockout of CSB in U2OS-265 cells (Supplementary Fig. 4a) resulted in an increase in accumulation of RIF1 and its effector MAD2L2[39, 40] at FokI-induced DSBs (Fig. 3c, d and Supplementary Fig. 4b). Loss of CSB did not affect GFP-PTIP recruitment to FokI-induced DSBs (Fig. 3e), suggesting that CSB specifically restricts the RIF1-MAD2L2 pathway but not the parallel PTIP pathway[41]. Knockdown of the TCR protein CSA had little impact on the accumulation of RIF1 and MAD2L2 at FokI-induced DSBs but sensitized cells to UV radiation (Supplementary Fig. 4c–f), suggesting that CSA is not involved in inhibition of the RIF1-MAD2L2 pathway. These results further imply that the function of CSB in DNA DSB repair pathway choice is unlikely to be a general feature of TCR proteins.

RIF1-MAD2L2 accumulation in CSB knockout cells was accompanied by an impairment in BRCA1 accumulation at FokI-induced DSBs (Fig. 3f, Supplementary 4g). In support of the observed impairment in BRCA1 accumulation, CSB null cells exhibited reduced DSB repair by HR but increased DSB repair by NHEJ (Fig. 3g, h). Introduction of Myc-CSB into U2OS-265 CSB knockout cells not only suppressed RIF1 but also restored BRCA1 accumulation at FokI-induced DSBs (Fig. 3i, j). On the other hand, neither Myc-CSB-GG nor Myc-CSB-GGG were able to dampen RIF1 and restore BRCA1 accumulation in U2OS-265 CSB knockout cells (Figs. 3i, j). These results suggest that the CSB WHD is necessary for regulating DSB pathway choice. These results further imply that CSB acts as an inhibitor of RIF1.

**RIF1 recruits CSB to DSBs in S phase**. Analysis of the dynamics of Myc-CSB accumulation at FokI-induced DSBs in synchronized U2OS-265 cells revealed that CSB recruitment to DSBs was cell cycle regulated, peaking in S phase (Fig. 4a). Cell synchronization did not significantly affect the induction of DSBs by FokI (Fig. 4b). At 0 h post release from a double thymidine block, about 6% of cells exhibited Myc-CSB accumulation at FokI-induced DSBs (Fig. 4a). This value increased sharply to about 30% at 2 h post release and peaked to about 38% at 6 h post release when the majority of cells (62.6%) were in S phase (Fig. 4a, Supplementary Fig. 5a). A dramatic decline in the number of cells exhibiting Myc-CSB accumulation was detected 16 h post release when the majority of cells (58.2%) were in G1 (Fig. 4a and Supplementary Fig. 5a),

RIF1 accumulation at FokI-induced DSBs was at the highest level in cells 0 h post release when the majority of cells were arrested in G1 (Fig. 4a, Supplementary Fig. 5a), in agreement with a previous finding that RIF1 is largely recruited to sites of DSBs in G1 cells[9]. RIF1 accumulation started to decline as cells entered S phase and continued to decline as cells progressed through S and G2, dipping to the lowest level at 10 h post release when cells were enriched in G2/M (Fig. 4a and Supplementary Fig. 5a). Despite this decline, a substantial number of cells retained RIF1 at FokI-induced DSBs in S/G2 phase, particularly from 2 to 6 h post release (45%, 44 and 38% at respective 2 h, 4 h, and 6 h post release) (Fig. 4a). These results prompted us to ask if this pool of RIF1 might be responsible for the sharp increase in CSB recruitment to FokI-induced DSBs observed earlier from 2 to 6 h post release. To address this question, we turned to 53BP1 knockout cells to avoid any potential replication defect associated with RIF1 deficiency[42]. Knockout of 53BP1 did not alter the cell cycle profile of U2OS-265 cells (Supplementary Fig. 5b) nor did it affect expression of RIF1 or CSB (Fig. 4c). While knockout of 53BP1 did not affect the induction of DSBs by FokI (Fig. 4d), it

completely abrogated RIF1 accumulation at FokI-induced DSBs (Fig. 4e). Recruitment of Myc-CSB to FokI-induced DSBs was also severely impaired in 53BP1 knockout U2OS-265 cells at 2, 4, and 6 h post release (Fig. 4f). These results suggest that RIF1 is responsible for recruiting CSB to DSBs in S phase.

**CSB inhibits RIF1 but promotes BRCA1 in S/G2**. Our earlier finding that CSB acts as an inhibitor of RIF1 prompted us to ask if this inhibition might occur in S/G2 phase. Loss of CSB did not affect the induction of DSBs by FokI (Fig. 4g) but prevented the decline in RIF1 accumulation at the FokI-induced DSBs in cells from 2 to 8 h post release from a double thymidine block (Fig. 4h). RIF1 accumulation at FokI-induced DSBs in CSB null cells was similar to that in wild-type cells at 12 h post release when cells started to exit G2/M and were enriched in G1 (Fig. 4h and Supplementary Fig. 5a), further supporting the notion that CSB inhibits RIF1 specifically in S/G2. The elevated accumulation of RIF1 in S/G2 was associated with a decrease in BRCA1 accumulation at FokI-induced DSBs (Fig. 4i), which was unlikely due to a loss of BRCA1 expression in CSB knockout cells (Fig. 4j). These results reveal that CSB inhibits RIF1 but promotes BRCA1 in S/G2.

**CSB evicts histones from chromatin flanking DNA DSBs in vivo**. To investigate if CSB might function as a chromatin remodeler at DSBs, we employed a well-established inducible ddI-PpoI expression construct[18], which was stably integrated into both hTERT-RPE wild-type and CSB knockout cells (Supplementary Fig. 6a). I-PpoI has a number of cleavage sites in the human genome[18], including a unique site on chromosome 1. Neither the ability of I-PpoI to induce DSBs nor the percentage of I-PpoI-induced cleavage on chromosome 1 was affected by loss of CSB in hTERT-RPE cells (Supplementary Fig. 6b–d).

ChIP analysis revealed that loss of both histone H2A and H2B from chromatin surrounding the unique I-PpoI cleavage site on chromosome 1 in hTERT-RPE wild-type cells was visible 1 h following I-PpoI induction and peaked two hours post I-PpoI induction (Figs. 5a, b), in agreement with previous observations that histones are removed from chromatin surrounding DSBs to accommodate HR-mediated repair[18, 43, 44]. On average, 45–50% of loss of H2A and H2B was observed 2 h post I-PpoI induction (Figs. 5a, b) and this effect was similar to that previously reported[16, 18]. At 2 h post induction, the average frequency of I-PpoI-induced cleavage was 21% (Supplementary Fig. 6d). Previously I-PpoI was reported to cleave this locus at a frequency of ~30% in MCF7 cells[18]. Perhaps the cleavage frequency by I-PpoI varies depending upon the cell type.

On the other hand, induction of I-PpoI did not lead to any significant removal of H2A and H2B from the I-PpoI cleavage site on chromosome 1 in hTERT-RPE CSB knockout cells (Fig. 5c, d). To further substantiate the role of CSB in removing H2A and H2B from damaged chromatin, we generated hTERT-RPE-ddIPpoI-CSB KO cells stably expressing the vector alone, Myc-CSB or Myc-CSB carrying a W851R ATPase-dead mutation[27, 30]. We then examined histone H2A and H2B occupancy in these cell lines 2 h post induction of I-PpoI expression when loss of H2A and H2B was observed earlier to peak in hTERT-RPE parental cells. This time point was also used in experiments below for analysis of other CSB mutant alleles. While wild-type CSB rescued I-PpoI-induced loss of H2A and H2B from the I-PpoI cleavage site on chromosome 1 in CSB knockout cells, the ATPase-dead mutant CSB-W851R failed to do so (Fig. 5e, f). These results demonstrate that CSB functions as a chromatin remodeler in vivo and that its ATP-dependent chromatin

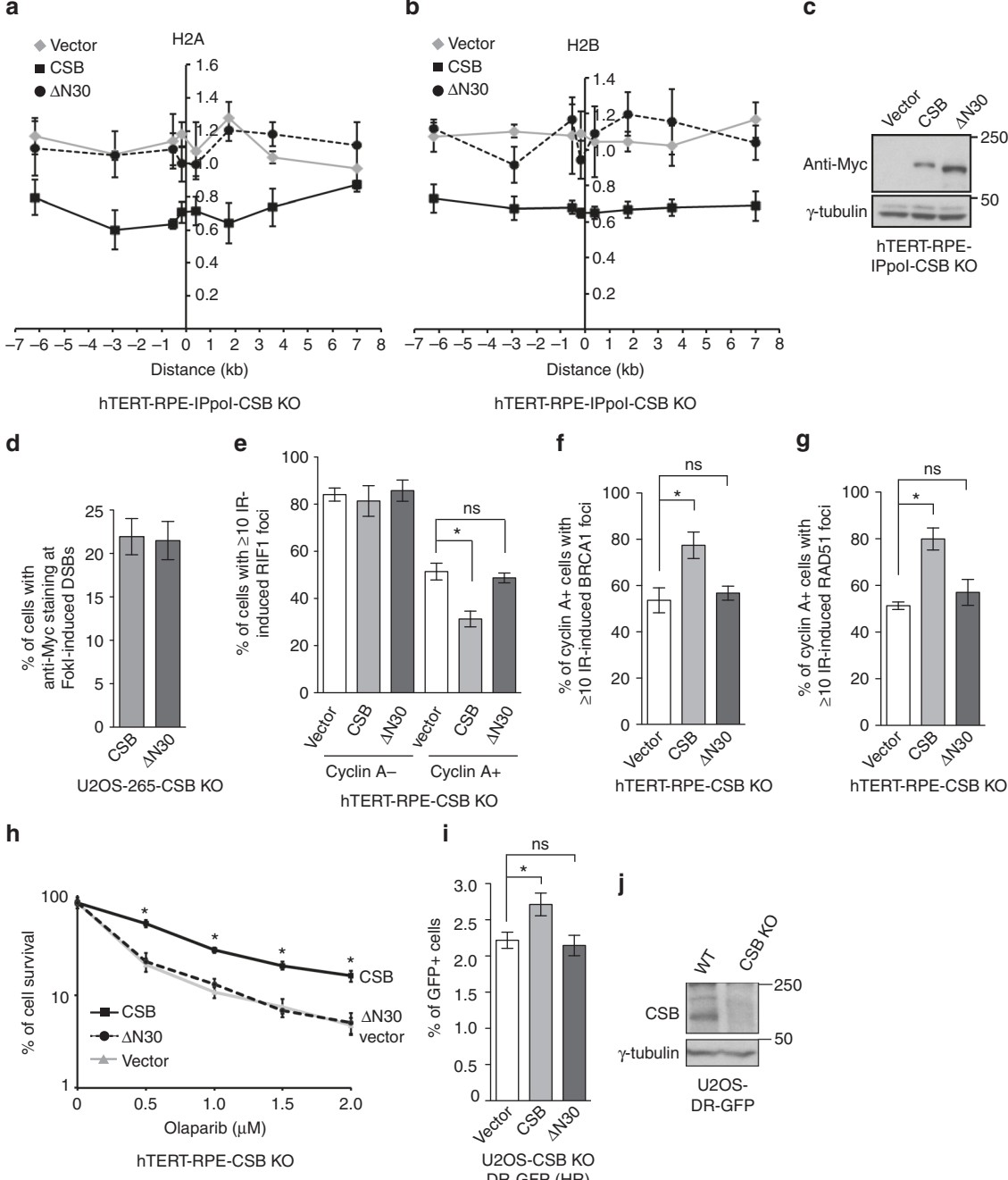

**Fig. 6** The N-terminus of CSB mediates its chromatin remodeling activity to repress RIF1 accumulation at sites of DSBs. **a** Relative occupancy of histone H2A in ddI-PpoI-expressing CSB KO hTERT-RPE cells complemented with various alleles as indicated. Both x- and y-axes are as described in Fig. 5a. SEM from three independent experiments are indicated. **b** Relative occupancy of histone H2B in ddI-PpoI-expressing CSB KO hTERT-RPE cells complemented with various alleles as indicated. Both x- and y-axes are as described in Fig. 5a. SEM from three independent experiments are indicated. **c** Western analysis of hTERT-RPE-IPpoI-CSB KO cells expressing various alleles as indicated. **d** Quantification of the percentage of Myc-CSB and Myc-CSB-ΔN30-expressing U2OS-265 cells exhibiting anti-Myc staining at FokI-induced DSBs. A total of 250 cells positive for anti-Myc staining were scored for each independent experiment in a blind manner. SDs from three independent experiments are indicated. **e** Quantification of the percentage of cyclin A- and cyclin A + cells with 10 or more IR-induced RIF1 foci. hTERT-RPE CSB KO cells stably expressing various alleles as indicated were treated with 2 Gy IR and fixed 1 h post IR. A total of 500–550 cells were scored for each independent experiment in a blind manner. SDs from three independent experiments are indicated. *$P < 0.05$. ns: $P > 0.05$ (Student t test). **f** Quantification of the percentage of cyclin A + cells with ≥ 10 IR-induced BRCA1 foci. Scoring was done as in **e**. SDs from three independent experiments are indicated. *$P < 0.05$. ns: $P > 0.05$ (Student t test). **g** Quantification of the percentage of cyclin A + cells with 10 or more IR-induced RAD51 foci. hTERT-RPE CSB KO cells stably expressing various alleles as indicated were treated with 2 Gy IR and fixed 4 h post IR. Scoring was done as in **e**. SDs from three independent experiments are indicated. *$P < 0.05$. ns: $P > 0.05$ (Student t test). **h** Clonogenic survival assays. SDs from three independent experiments are indicated. *$P < 0.05$ (Student t test) for comparison between CSB and ΔN30. **i** HR-mediated repair. SDs from three independent experiments are indicated. *$P < 0.05$. ns: $P > 0.05$ (Student t test). **j** Western analysis of U2OS-DR-GFP parental and CSB KO cells

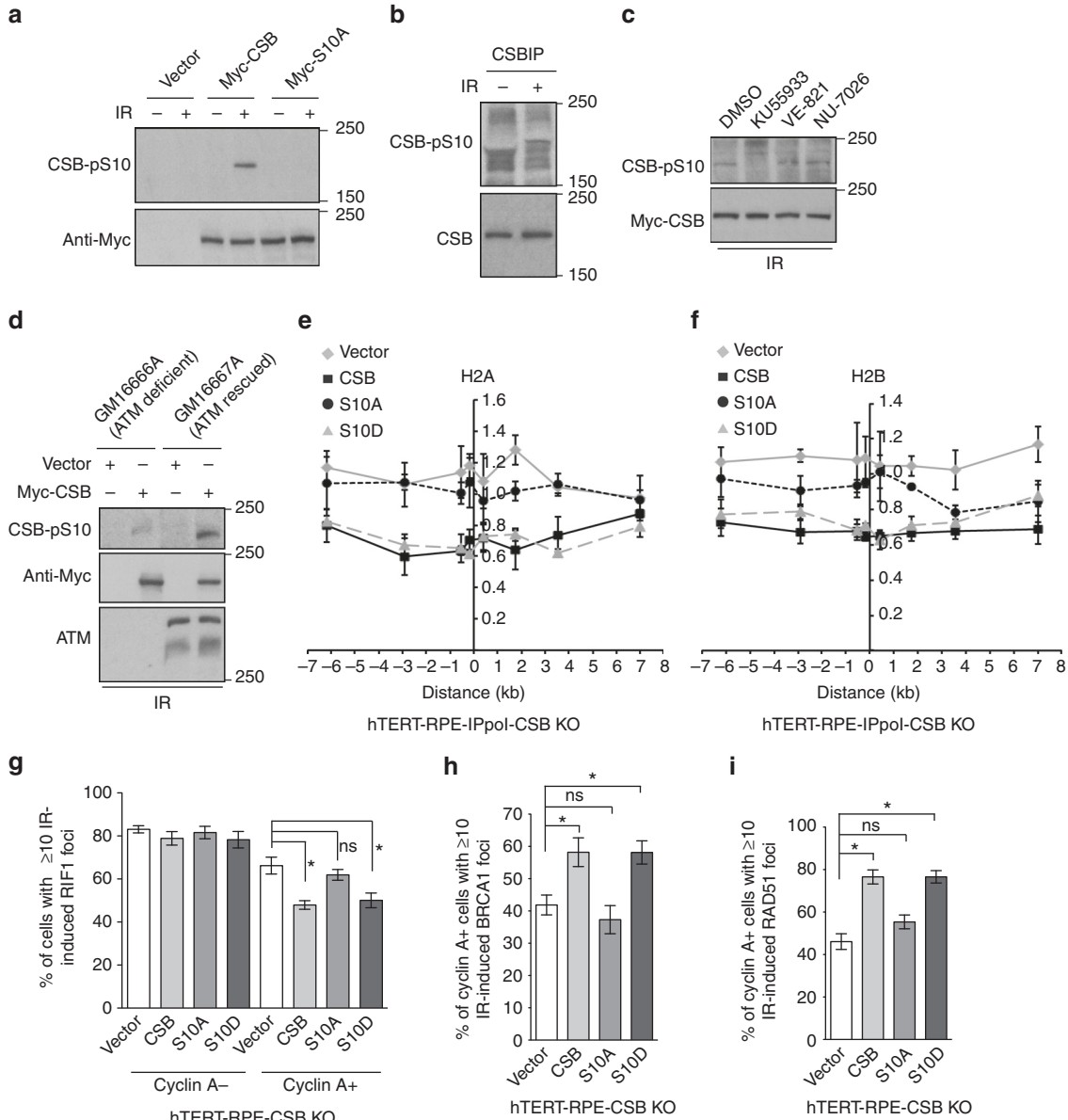

**Fig. 7** ATM controls the chromatin remodeling activity of CSB through S10 phosphorylation. **a** Western analysis of U2OS CSB KO cells stably expressing various alleles as indicated. Cells were either treated with or without 10 Gy IR. **b** Western analysis of anti-CSB immunoprecipitates from HCT116 cells treated with or without 10 Gy IR. **c** Western analysis. Myc-CSB-expressing U2OS CSB KO cells were treated with DMSO, ATM inhibitor KU55933, ATR inhibitor VE-821 or DNA-PKcs inhibitor NU-7026 for 1 h prior to 10 Gy IR. **d** Western analysis of ATM-deficient GM16666A and ATM-complemented GM16667A cells. Cells were transfected with the vector alone or Myc-CSB, followed by treatment with 10 Gy IR 48 h post transfection. **e** Relative occupancy of histone H2A in ddI-Ppol-expressing hTERT-RPE CSB KO cells complemented with various alleles as indicated. Both x- and y-axes are as described in Fig. 5a. SEM from three independent experiments are indicated. **f** Relative occupancy of histone H2B in ddI-Ppol-expressing hTERT-RPE CSB KO cells complemented with various alleles as indicated. Both x- and y-axes are as described in Fig. 5a. SEM from three independent experiments are indicated. **g** Quantification of the percentage of cyclin A- and cyclin A+ cells with 10 or more IR-induced RIF1 foci. hTERT-RPE CSB KO cells stably expressing various alleles as indicated were treated with 2 Gy IR and fixed 1 h post IR. A total of 500–550 cells were scored for each independent experiment in a blind manner. SDs from three independent experiments are indicated. $*P < 0.05$. ns: $P > 0.05$ (Student t test). **h** Quantification of the percentage of cyclin A+ cells with $\geq 10$ IR-induced BRCA1 foci. Scoring was done as in 7 g. SDs from three independent experiments are indicated. $*P < 0.05$. ns: $P > 0.05$ (Student t test). **i** Quantification of the percentage of cyclin A+ cells with 10 or more IR-induced RAD51 foci. Cells were treated with 2 Gy IR and fixed 4 h post IR. Scoring was done as in 7 g. SDs from three independent experiments are indicated. $*P < 0.05$. ns: $P > 0.05$ (Student t test)

remodeling activity is essential for displacing histones from chromatin flanking DSBs.

Earlier, we have shown that loss of CSB impairs BRCA1 accumulation at FokI-induced DSBs (Fig. 3f). BRCA1 is reported to mediate H2A ubiquitylation that is recognized by the ubiquitin-binding CUE domain of chromatin remodeler SMARCAD1[13]. Formally it was possible that loss of histone displacement in CSB null cells might have resulted from impaired recruitment of SMARCAD1 at DSBs. However, we did not detect any significant change in SMARCAD1 recruitment to FokI-induced DSBs in CSB null cells (Supplementary Fig. 7a, b). Combined with previous reports that CSB is a chromatin remodeler in vitro[30, 32, 33], our finding supports the notion that CSB functions as a chromatin remodeler in vivo. Our finding

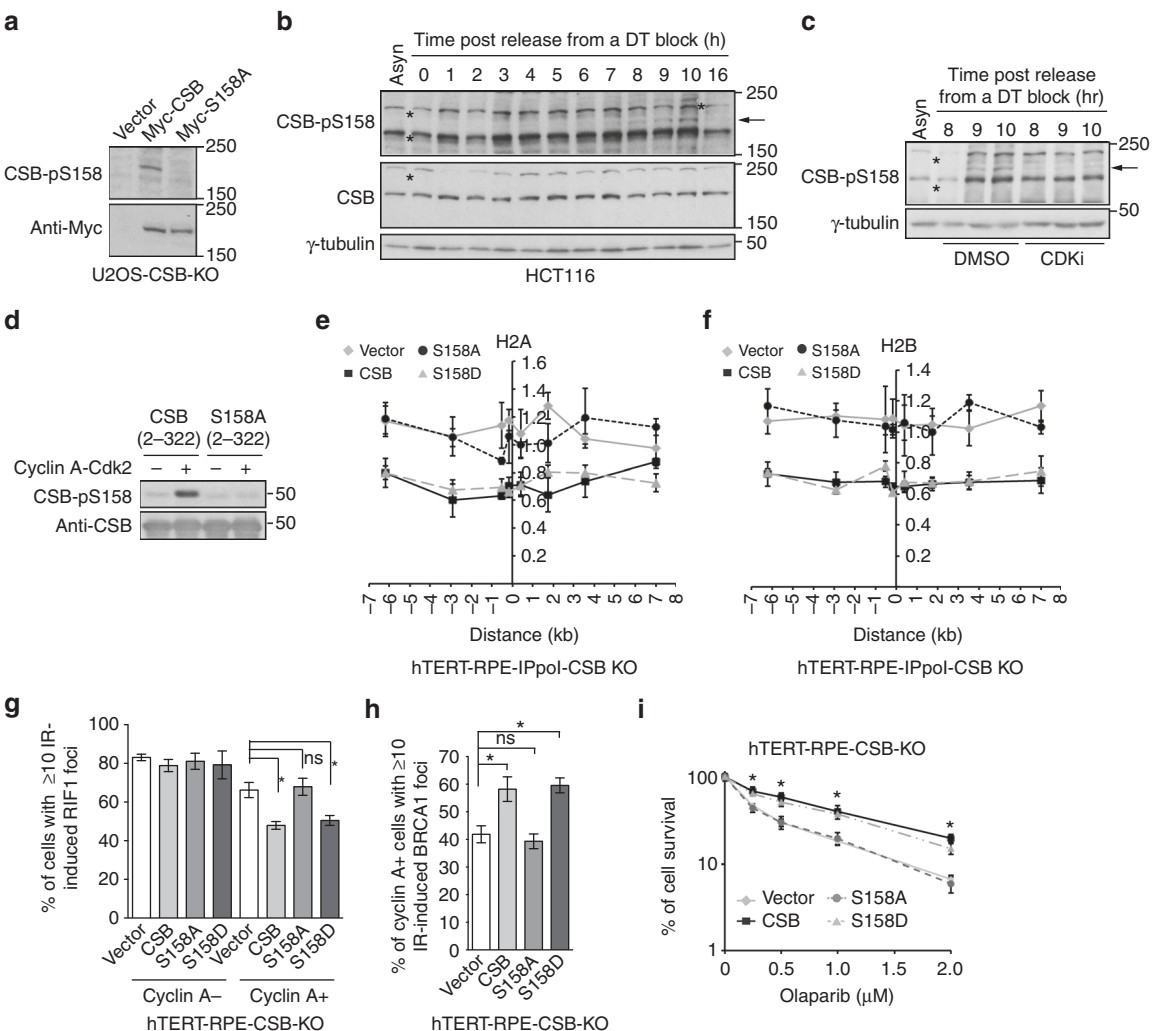

**Fig. 8** Cyclin A–CDK2 controls the chromatin remodeling activity of CSB through S158 phosphorylation. **a** Western analysis of U2OS-CSB-KO cells stably expressing the vector alone, Myc-CSB or Myc-CSB carrying a S158A mutation. Immunoblotting was performed with anti-CSB-pS158 and anti-Myc antibodies. **b** Western analysis of synchronized HCT116 cells. The arrow indicates the position of CSB-pS158. Asterisks indicate non-specific bands. **c** Western analysis. Asynchronous and synchronized HCT116 cells post release from a double thymidine block as indicated were treated with DMSO or the CDK inhibitor roscovitine. The arrow indicates the position of CSB-pS158. Asterisks indicate non-specific bands. **d** In vitro kinase assays with recombinant cyclin A-CDK2 and bacteria-expressed recombinant CSB fragments as indicated. **e** Relative occupancy of histone H2A in ddI-PpoI-expressing hTERT-RPE CSB KO cells complemented with various alleles as indicated. Both x- and y-axes are as described in Fig. 5a. SEM from three independent experiments are indicated. **f** Relative occupancy of histone H2B in ddI-PpoI-expressing hTERT-RPE CSB KO cells complemented with various alleles as indicated. Both x- and y-axes are as described in Fig. 5a. SEM from three independent experiments are indicated. **g** Quantification of the percentage of cyclin A− and cyclin A+ cells with 10 or more IR-induced RIF1 foci. hTERT-RPE CSB KO cells stably expressing various alleles as indicated were treated with 2 Gy IR and fixed 1 h post IR. A total of 500–550 cells were scored for each independent experiment in a blind manner. SDs from three independent experiments are indicated. *$P < 0.05$. ns: $P > 0.05$ (Student t test). **h** Quantification of the percentage of cyclin A+ cells with ≥ 10 IR-induced BRCA1 foci. Scoring was done as in 8 g. SDs from three independent experiments are indicated. *$P < 0.05$. ns: $P > 0.05$ (Student t test). **i** Clonogenic survival assays of olaparib-treated hTERT-RPE CSB-KO cells complemented with various alleles as indicated. SDs from three independent experiments are indicated. *$P < 0.05$ (Student t test) for comparison between CSB and S158A

further implies that CSB may act independently of SMARCAD1 in promoting HR-mediated DSB repair.

**Chromatin remodeling by CSB N-terminus inhibits RIF1 at DSBs.** Deletion analysis revealed that deleting the first 30 amino acids from CSB N-terminus (CSB-ΔN30) was sufficient to abrogate its ability to displace H2A and H2B from the I-PpoI cleavage site on chromosome 1 in hTERT-RPE-ddIPpoI-CSB KO cells (Fig. 6a, b). This inability was unlikely due to a lack of expression or a defect in recruitment of CSB-ΔN30 to DSBs (Fig. 6c, d). These results suggest that CSB N-terminus is necessary for its in vivo chromatin remodeling activity at DSBs.

Previously it has been reported that CSB limits IR-induced RIF1 foci formation specifically in S/G2 cells[27]. When stably introduced into hTERT-RPE CSB null cells, Myc-CSB-ΔN30 failed to fully suppress IR-induced RIF1 foci formation in cells staining positive for cyclin A, a marker for S/G2 cells (Fig. 6e). The inability of Myc-CSB-ΔN30 to suppress RIF1 foci formation was accompanied by a lack of rescue in IR-induced BRCA1 and RAD51 foci formation (Fig. 6f, g). Overexpression of Myc-CSB-ΔN30 failed to suppress the sensitivity of hTERT-RPE CSB null cells, which were proficient for both BRCA1[27] and BRCA2 (N.L. Batenburg and X.D. Zhu, unpublished data), to olaparib treatment (Fig. 6h). Myc-CSB-ΔN30 also failed to promote HR

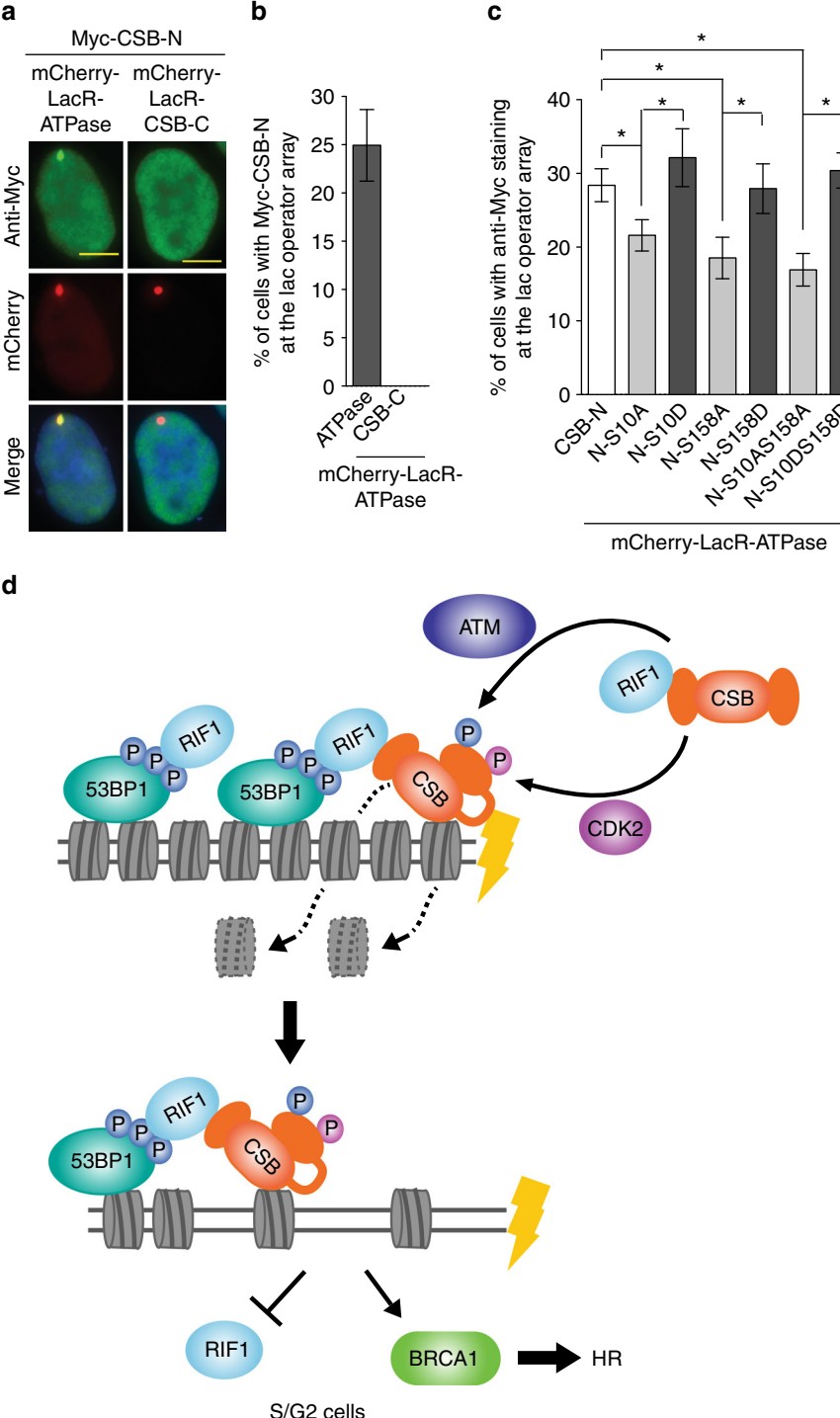

**Fig. 9** CSB phosphorylations on S10 and S158 modulate the interaction of the N-terminus with the ATPase domain. **a** Immunofluorescence of U2OS-265 cells transfected with Myc-CSB-N in conjunction with either mCherry-LacR-CSB-ATPase or mCherry-LacR-CSB-C. Scale bars, 5 μm. **b** Quantification of the percentage of U2OS-265 cells from **a** with anti-Myc staining at FokI-induced DSBs. A total of 250 Myc-expressing cells were scored for each independent experiment in a blind manner. SDs from three independent experiments are indicated. **c** Quantification of the percentage of U2OS-265 cells with anti-Myc staining at FokI-induced DSBs. U2OS-265 were co-transfected with various alleles as indicated. A total of 250 Myc-expressing cells were scored for each independent experiment in a blind manner. SDs from three independent experiments are indicated. *$P < 0.05$ (Student $t$ test). **d** Model for control of CSB chromatin remodeling activity by ATM and CDK2 in DNA DSB repair pathway choice in S/G2. See the text for details

in the reporter U2OS-DR-GFP CSB KO cells (Fig. 6i, j). These results suggest that the chromatin remodeling activity of CSB is necessary to suppress RIF1 but promote BRCA1-mediated HR in S/G2.

**ATM controls the chromatin remodeling activity of CSB.** Analysis of the first 30 amino acids of CSB revealed three closely-spaced SQ/TQ motifs ($S^{10}Q$, $T^{12}Q$ and $S^{20}Q$), which are commonly found in DNA-damage response proteins that are

substrates of ATM/ATR[45]. Clonogenic survival assays revealed that although Myc-CSB carrying a nonphosphorylatable mutation of either T12A or S20A behaved like wild-type CSB in suppressing the sensitivity of CSB null cells to olaparib (Supplementary Fig. 8a, b), Myc-CSB carrying a nonphosphorylatable mutation of S10A failed to suppress the sensitivity of CSB null cells to olaparib (Supplementary Fig. 8c). On the other hand, Myc-CSB carrying a phosphomimic mutation of S10D was fully competent in suppressing the sensitivity of CSB null cells to olaparib (Supplementary Fig. 8c). All CSB mutants were expressed at a level comparable to wild-type CSB (Supplementary Fig. 8d). These results suggest that CSB phosphorylation on S10 is important for its function in DSB repair.

Western analysis with an antibody raised against a peptide carrying phosphorylated S10 revealed that both endogenous CSB and Myc-CSB were phosphorylated on S10 following induction of IR-induced DNA damage and that little CSB-pS10 was detected in undamaged cells (Fig. 7a, b). CSB phosphorylation on S10 was sensitive to the ATM inhibitor KU55933, but not the ATR inhibitor VE-821 or the DNA-PKcs inhibitor NU7026 (Fig. 7c). Introduction of wild-type ATM into ATM-deficient GM16666A cells rescued IR-induced CSB phosphorylation on S10 (Fig. 7d). Together, these results reveal that ATM is the main kinase responsible for damage-induced CSB phosphorylation on S10. ATM-deficient GM16666A cells carry a homozygous frameshift mutation at codon 762 of the *ATM* gene and no ATM protein is detected in these cells[46] (Fig. 7d). The residual signal of CSB-pS10 observed in GM16666A cells might result from an activity of another kinase in the absence of ATM.

ChIP analysis revealed that following induction of I-PpoI, Myc-CSB-S10A failed to displace H2A and H2B from the I-PpoI cleavage site on chromosome 1, whereas Myc-CSB-S10D was able to do so (Fig. 7e, f). The inability of Myc-CSB-S10A to evict H2A and H2B was unlikely due to a lack of protein expression (Supplementary Fig. 8e). Both Myc-CSB-S10A and Myc-CSB-S10D were recruited to FokI-induced DSBs, indistinguishable from Myc-CSB (Supplementary Fig. 8f). These results suggest that ATM controls the chromatin remodeling activity of CSB at DSBs through damage-induced S10 phosphorylation.

When stably introduced into hTERT-RPE CSB KO cells, overexpression of Myc-CSB-S10A failed to suppress IR-induced RIF1 foci formation in cyclin A-positive cells, whereas overexpression of Myc-CSB-S10D was able to do so (Fig. 7g). IR-induced BRCA1 and RAD51 foci formation was compromised in CSB null cells complemented with Myc-CSB-S10A but not in CSB null cells complemented with Myc-CSB-S10D (Fig. 7h, i). These results suggest that ATM-dependent chromatin remodeling activity of CSB is part of the mechanism that suppresses RIF1 but promotes BRCA1 and RAD51 at DSBs in S/G2 cells.

**CDK2 controls the chromatin remodeling activity of CSB.** Mass spectrometric analysis of Flag-CSB immunoprecipitated from IR-treated cells revealed a robust phosphorylation of S158 (S[158]P) (Supplementary Fig. 9a), which fits the consensus sequence (S/TP) for cyclin-dependent kinases. Western analysis with an antibody raised against phosphorylated S158 confirmed that S158 was phosphorylated in vivo (Fig. 8a, b). Analysis of synchronized cell lysates revealed that CSB phosphorylation on S158, which was absent in CSB null cells (Supplementary Fig. 9d), was reproducibly detected above the background level at 6 h post release from a double thymidine block, continued to increase as cells progressed through S/G2/M but declined sharply when cells returned to G1, 16 h post release (Fig. 8b and Supplementary Fig. 9b, c). Treatment with the CDK inhibitor roscovitine severely affected S158 phosphorylation (Fig. 8c). Furthermore, S158 was

an in vitro substrate of cyclin A-CDK2 (Fig. 8d). Together, these results suggest that cyclin A-CDK2 is a kinase responsible for CSB phosphorylation on S158 in S/G2 phase.

ChIP analysis revealed that Myc-CSB carrying a nonphosphorylatable mutation of S158A failed to rescue the displacement of H2A and H2B from the I-PpoI cleavage site on chromosome 1 in CSB null cells whereas Myc-CSB carrying a phosphomimic mutation of S158D was fully competent to do so (Fig. 8e, f). The level of CSB-S158A expression was comparable to that of wild-type CSB and CSB-S158D (Supplementary Fig. 9e). Both Myc-CSB-S158A and Myc-CSB-S158D were recruited to FokI-induced DSBs, indistinguishable from Myc-CSB (Supplementary Fig. 9f). These results suggest that CSB phosphorylation on S158 by cyclin A-CDK2 controls its chromatin remodeling at DSBs in S/G2 cells.

When stably introduced into hTERT-RPE CSB KO cells (Supplementary Fig. 9g), Myc-CSB-S158A failed to suppress IR-induced RIF1 foci formation in cyclin A-positive cells (Fig. 8g), failed to rescue IR-induced BRCA1 and RAD51 foci formation and failed to support cell survival in response to olaparib treatment (Fig. 8h, i and Supplementary Fig. 9h). Myc-CSB-S158A also failed to promote HR in U2OS-DR-GFP CSB KO cells (Supplementary Fig. 9i). On the other hand, Myc-CSB-S158D was fully competent in suppressing IR-induced RIF1 foci formation in cyclin A-positive CSB null cells and facilitating efficient HR as evidenced by a complete rescue in IR-induced BRCA1 and RAD51 foci formation, HR-mediated repair as well as cell survival of CSB null cells in response to olaparib treatment (Fig. 8h, i and Supplementary Fig. 9h, i). Collectively, these results suggest that cyclin A-CDK2 controls the chromatin remodeling activity of CSB at DSBs to promote efficient HR.

**Phosphorylation controls CSB intramolecular interaction.** We observed a robust interaction of Myc-CSB-N with mCherry-LacR-CSB-ATPase at the lac operator array (Fig. 9a, b), in agreement with previous reports that CSB N-terminal region is engaged in an intramolecular interaction with the ATPase domain[30, 31]. No interaction of Myc-CSB-N with mCherry-LacR-CSB-C was detected (Fig. 9a, b). We found that Myc-CSB-N interaction with mCherry-LacR-CSB-ATPase was impaired by a single mutation of either S10A or S158A but not by a single mutation of either S10D or S158D (Fig. 9c). Further analysis of double mutations of either S10AS158A or S10DS158D revealed that CSB phosphorylations on S10 and S158 acted in the same epistatic pathway to promote the interaction of the N-terminal region with the ATPase domain. These results imply that these two phosphorylation events serve as molecular gates to modulate intramolecular interactions of CSB N-terminal region with the ATPase domain.

**Discussion**

The work presented here has not only uncovered that CSB interacts with RIF1 and is recruited by RIF1 to DSBs in S phase but also provided the first direct evidence that CSB functions as a chromatin remodeler in vivo. Our data suggest that CSB phosphorylations on S10 by ATM and on S158 by cyclin A-Cdk2 serve as molecular signals governing its chromatin remodeling activity at DSBs, which inhibits RIF1 but promotes BRCA1-mediated HR (Fig. 9d).

Our finding that RIF1 can form a subcomplex with CSB independently of 53BP1 and damage induction suggests that CSB may be recruited to DSBs in the form of this subcomplex via RIF1 interaction with 53BP1 (Fig. 9d). However, we cannot rule out the possibility that CSB may be recruited to DSBs via the

RIF1–53BP1 complex. Previously it has been reported that CSB recruitment to DSBs is dependent upon transcription[27]. RNA is reported to mediate 53BP1 and RIF1 recruitment to DSBs[47, 48]. Perhaps, CSB recruitment by RIF1 to DSBs might also be regulated by transcription, which would require future investigation.

We have shown that RIF1 interacts with CSB through its conserved CTD. The CTD of RIF1 has been implicated in binding BLM to promote recovery of stalled replication forks[49]. Knockdown of BLM did not affect CSB recruitment to FokI-induced DSBs (N.L. Batenburg and X.D. Zhu, unpublished data), suggesting that it is unlikely that RIF1 mediates CSB recruitment to DSBs through BLM.

We have uncovered that the very C-terminus of CSB harbors a cryptic winged helix domain (WHD), which is predicted to be evolutionarily conserved from yeast to human and shares closest resemblance to the WHD of general transcription factors (RAP70, RAP30, TAF1, ELL and ELL2) and the chromatin assembly factor CAF1. The WHD is a versatile domain that is implicated in protein-DNA and protein–protein interactions[50]. Recent studies suggest that the last 30 amino acids of CSB is necessary for its interaction with RNAPII in transcription-coupled UV repair[51]. Our finding that CSB interacts with RIF1 through its WHD in DSB repair supports the notion that the CSB WHD acts as a protein–protein interaction module to mediate its interaction with different partners depending upon the type of DNA repair process. The CSB WHD is predicated to span amino acids 1417–1493 and overlaps with the previously reported ubiquitin-binding domain (UBD) of CSB[36]. Computer modeling suggests that the two leucines 1427 and 1428, which have previously been implicated in the function of the UBD in UV repair[36], are contained within helix 1 of the WHD. Further structural and functional analysis is needed to clarify the role of this region in regulating CSB activity.

We have shown that CSB N-terminal region interacts with its ATPase domain and that this interaction is modulated by two CSB phosphorylation events on its N-terminal S10 and S158, both of which are necessary for its in vivo chromatin remodeling activity at DSBs. We envision a model in which in the absence of DSBs, CSB N-terminal region interacts with its ATPase domain in such a manner that its ATPase activity is restricted. Upon induction of DSBs, ATM- and CDK2-dependent CSB phosphorylation of S10 and S158 promotes CSB conformational change, which releases the inhibitory effect of its N-terminal region on its ATPase activity (Fig. 9d). In the absence of these two phosphorylation events, CSB-ΔN30, CSB-S10A and CSB-S158A mutants would not be able to undergo protein conformational change needed for stimulation of its ATPase activity. Our finding that CSB phosphorylations on S10 and S158 stimulate CSB-N interaction with CSB-ATPase suggests that these two phosphorylation events might create an interface favorable for these two domains to interact. We propose that these two phosphorylation events act together as molecular signals to trigger the release of the autoinhibition of its N-terminal region on its ATPase domain in S/G2 cells (Fig. 9d). Subsequently the chromatin remodeling activity of CSB at DSBs evicts histones and disassembles nucleosomes, limiting RIF1 accumulation but paving the way for BRCA1-mediated HR activity in S/G2 (Fig. 9d). Our finding that the activation of CSB chromatin remodeling activity at DSBs requires not only a DNA-damage signal but also a signal indicating the correct phase of the cell cycle suggests that these two signals are needed to restrict displacement of histones by CSB to damaged S/G2 cells, perhaps helping guard against unwarranted extensive chromatin disassembly by CSB in undamaged cells or damaged G1 cells.

## Methods

**Plasmids, siRNA and antibodies.** Retroviral expression constructs for wild-type CSB and ATPase-dead mutant CSB-W851R have been described[26, 27]. Wild-type CSB was used as a template to generate various CSB deletion alleles, which were cloned into the retroviral expression vector pLPC-NMyc[26], mammalian expression vector mCherry-LacR-NLS[9] or the bacterial expression pHis-parallel-2[52]. The QuickChange site-directed mutagenesis kit (Agilent Technologies) was used to generate CSB mutants S10A, S10D, S158A and S158D. The primers used to clone CSB deletions and point mutations are available upon request. To generate pBabe-neo-ddI-PpoI expression construct, pBabe-ddI-PpoI[18] (#49052, Addgene) was digested with *Bam*HI and *Sal*I and two inserts (a 267-bp *Bam*HI-*Sal*I fragment and a 1.5–kb *Bam*HI fragment) were sequentially ligated into BamHI-SalI-linearized pBabe-Neo (a kind gift from Titia de Lange, Rockefeller University). Inserts of all plasmids was confirmed by DNA sequencing.

siRNAs against RIF1 and CSA were obtained from Dhamacon (siRIF1, D–027983–02–0005; siCSA, J-011008–07–0002). The expression constructs (pDEST-mCherry-LacR and pDEST-EGFP) carrying either siRIF1-resistant wild-type RIF1 or various RIF1 deletion alleles have been described in ref. [9]. The GFP-PTIP expression construct[41] was a generous gift from André Nussenzweig and Jeremy Daniel.

Rabbit polyclonal anti-pS10 and anti-pS158 antibodies were developed by Cocalico Biologicals against respective CSB peptides containing phosphorylated serine 10 (NEGIPHS-pS-QTQEQDC) (Bio-Synthesis Inc) and phosphorylated serine 158 (NKIIEQL-pS-PQAATSR) (Bio-Synthesis Inc). Other antibodies used were listed in Supplementary Table 1.

**Cell culture and drug treatment.** All cells were grown in DMEM medium with 10% fetal bovine serum supplemented with non-essential amino acids, L-glutamine, 100 U ml$^{-1}$ penicilin and 0.1 mg ml$^{-1}$ streptomycin. Cell lines used: hTERT-RPE parental and CSB knockout[27], Phoenix[26] (a kind gift from Titia de Lange, Rockefeller University), U2OS[53] (ATCC), U2OS-265[34] (a kind gift from Roger Greenberg, University of Pennsylvania), U2OS-DR-GFP[9], HCT116[54] (Life Technology), GM16666A[55, 56] (Coriell) and GM166667[55, 56] (Coriell). Parental cells were tested for mycoplasma contamination and were authenticated by STR DNA profiling. Retroviral gene delivery was carried out as described in ref. [57, 58] to generate stable cell lines. DNA and siRNA transfections were carried out with respective JetPRIME® transfection reagent (Polyplus) and Lipofectamine RNAiMax (Invitrogen) according to their respective manufacturer's instructions.

To induce expression of FokI, U2OS-265 cells were treated with both 1 μM Shield-1 (CheminPharma) and 4-hydroxytesterone (4-OHT, Abcam) for 6 h or for the indicated time. IR was delivered from a Cs-137 source at McMaster University (Gammacell 1000). Roscovitine (20 μM, Sigma),

KU55933 (10 μM, Sigma), VE-821 (10 μM, Selleckchem), NU7026 (10 μM, Sigma) were used to inhibit CDK, ATM, ATR and DNA-PKcs respectively.

**Mass spectrometric analysis of phosphorylated CSB.** Approximately 12 million U2OS cells stably expressing Flag-tagged CSB were treated with 20 Gy IR and the whole-cell extracts were prepared as described[59, 60]. Flag-CSB was immunoprecipitated from whole-cell extracts of approximately 12 million cells as described[59]. Affinity purification of Flag-CSB was carried out with FLAG® purification kit (Sigma) according to the manufacturer's instruction. Following the final wash in 50 mM ammonium bicarbonate (ABC), pH 8.0, the resin containing Flag-CSB was digested with 1 μg trypsin in 200 μl ABC buffer overnight at 37 ºC. The next day, a fresh 0.5 μg trypsin was added and the mixture was incubated for another 3 h. Following centrifugation, the supernatant was transferred to a keratin-free tube and fully dried. The dried peptides were reconstituted in 2% formic acid and diluted 1:5 with lactic acid solution [25% lactic acid, 60% acetonitrile (ACN), 2.5% trifluoroacetic acid (TFA)]. Phosphorylated peptides were enriched on titanium-oxide (Ti-O$_2$) tips (GL Sciences, Tokyo, Japan) that were equilibrated consecutively with 100% H$_2$O, 100% methanol and lactic acid solution. Following loading of the sample, tips were washed consecutively with lactic acid solution, 80% ACN plus 0.1% TFA, 0.1% TFA and 2× H$_2$O. Phosphorylated peptides were eluted with 400 mM NH$_4$OH. Peptides were dried and reconstituted in 5% formic acid and loaded onto a fused silica 12 cm analytical column packed in-house with 3.5 μm Zorbax C18 material (Agilent Technology). Peptides were analyzed using an Orbitrap ELITE (Thermo Scientific) coupled to an Eksigent nanoLC ultra (AB SCIEX). Peptides were eluted from the column using a 90 min period cycle with a linear gradient from 2 to 35% ACN in 0.1% formic acid. Tandem MS spectra were acquired in a data-dependent mode for the top 10 most abundant ions using collision induced dissociation. Acquired spectra were searched against the human Refseq_V53 database using Mascot.

**CRISPR/Cas9 genome editing of CSB.** U2OS, U2OS-265, U2OS-DR-GFP and HCT116 cells were transiently transfected with sgRNA (5′-AGAATTGC-CACTCTGAACGG-3′)[54] targeting CSB and expressed from the pX459 vector[61] (#48139, Addgene) containing Cas9 followed by the 2A-Puromycin cassette. The next day, cells were selected with puromycin for 2 days and subcloned to allow for

the formation of single colonies. Individual clones were screened by immuno-fluorescence with anti-CSB antibody (Fitzgerald) for the loss of CSB. CSB null clones were further confirmed by immunoblotting using anti-CSB antibody (Bethyl). Subsequently, any off-target effects from sgRNA were ruled out by clonogenic UV survival assays of CSB null clones complemented with either vector alone or Myc-tagged CSB. Only CSB null clones whose UV sensitivity were fully suppressed by wild-type CSB were used in this study.

**Cell synchronization and FACS analysis.** Cell synchronization was done essentially as described[53, 60] with some modifications. Cells were treated with 2 mM thymidine for 16 h, followed by washing in PBS three times and then released into fresh media for 9 h. Subsequently, cells were arrested again with 2 mM thymidine for 16 h and washed in PBS for three hours before their release into fresh media for various time points as indicated. For cell cycle analysis, two million cells from each of indicated time points were fixed and processed as described[53]. FACS analysis was performed on a FACSCalibur instrument and analyzed using FlowJo (v10.2). For induction of FokI expression in synchronized U2OS-265 cells, Shield-1 and 4-OHT were added 2 h prior to a given time point as indicated.

U2OS-DR-GFP WT and CSB knockout cells were transfected with indicated constructs along with an *I-SceI*-expressing plasmid using JetPRIME® transfection reagents (Polyplus). U2OS WT and CSB-KO cells were co-transfected with pEGFP-Pem1-Ad2 and *I-SceI* expression constructs. 48 h post transfection, cells were collected, fixed and subjected to FACS analysis as described[27]. A total of 10,000 cells per cell line were scored for each independent experiment. FACS analysis was performed on a FACSCalibur instrument.

**Immunoprecipitation and immunoblotting.** Immunoprecipitation (IP) with endogenous proteins was carried out as described[14] with minor modifications. Untreated HCT116 cells or HCT116 cells collected 1 h post 20 Gy IR were lysed in NETN buffer [20 mM Tris-HCl, pH 8.0, 100 mM NaCl, 1 mM EDTA, 0.5% Nonidet™ P-40 Substitute (Sigma), 1 mM PMSF, 1 µg ml⁻¹ aprotinin, 1 µg ml⁻¹ leupeptin, 1 µg ml⁻¹ pepstatin, 1 mM NaF, 1 mM NaVO₄, 50 mM Na-β-glycer-olphosphate] on ice for 30 min. For each IP, 5 mg of cell lysate was precleared with 30 µl protein G sepharose beads (GE Healthcare) for 1 h at 4 °C, followed by incubation with primary antibody (1–2 µg) overnight at 4 °C. Precipitates were then washed 4 times in NETN buffer containing 300 mM NaCl, and immuno-blotted with indicated antibodies. IP with an anti-Myc antibody in 293T cells co-overexpressed Myc-CSB-C and various mCherry-LacR-RIF1-CTD alleles was done as described[26]. Immunoblotting was performed as described[27].

**Chromatin immunoprecipitation (ChIP).** ChIP and I-PpoI-induced DSB assays were carried out as described[18, 62] with minor modifications. Cells stably expressing pBabe-neo-ddI-PpoI were first treated with Shield-1 (1 µM) for 3 h and then with 4-OHT (1 µM) for 15 min. Following fixation in 1% PBS-buffered formaldehyde for 10 min, cells were resuspended in 20× cell pellet volume of cell lysis buffer I [10 mM HEPES pH 6.5, 10 mM EDTA, 0.5 mM EGTA, 0.25% Triton X-100, 1 mM PMSF, 1 µg ml⁻¹ aprotinin, 1 µg ml⁻¹ leupeptin, 1 µg ml⁻¹ pepstatin, 1 mM NaF, 1 mM NaVO₄, 50 mM Na-β-glycerolphosphate] and incubated on ice for 10 min. Following centrifugation, cell pellets were washed in cell lysis buffer II [10 mM HEPES pH 6.5, 1 mM EDTA, 0.5 mM EGTA, 200 mM NaCl] and then resus-pended in nuclei lysis buffer [50 mM Tris-HCl pH8.1, 10 mM EDTA, 0.5% SDS]. Both cell lysis buffer II and nuclei lysis buffer contained phosphatase and protease inhibitors as described in cell lysis buffer I. Following incubation on ice for 10 min, the cell lysate was sonicated and clarified through centrifugation.

For each ChIP, 200 µl of the cell lysate was diluted 1:5 in IP dilution buffer [1% Triton X-100, 2 mM EDTA, 20 mM Tris-HCl pH8.1, 150 mM NaCl]. Out of 1 ml diluted lysate, 20 µl was set aside as input control and the remaining was precleared with protein G sepharose beads (GE Healthcare) preblocked with BSA and tRNA and then incubated with primary antibody (1 µg) overnight at 4 °C. Precipitates were washed once in low salt buffer [150 mM NaCl, 0.1% SDS, 1% Triton X-100, 2 mM EDTA, 20 mM Tris-HCl pH8.0, 1 mM PMSF, 1 µg ml⁻¹ aprotinin, 1 µg ml⁻¹ leupeptin, 1 µg ml⁻¹ pepstatin], once in high salt buffer [500 mM NaCl, 0.1% SDS, 1% Triton X-100, 2 mM EDTA, 20 mM Tris-HCl pH8.0], twice in LiCl buffer [0.25 M LiCl, 1% Nonidet™ P-40 Substitute, 1% deoxycholic acid, 1 mM EDTA, 10 mM Tris-HCl pH 8.0] and then once in TE buffer [10 mM Tris-HCl pH 8.0, 1 mM EDTA]. The IP DNA was eluted twice in elution buffer [0.1 M NaHCO₃, 1% SDS] at 65 °C for 15 min. Subsequently, the IP DNA, along with the input DNA (equivalent to 2% of lysate used for IP), were treated with RNase A at 37 °C for 1 h and then with proteinase K at 55 °C for 1 h. Following incubation overnight at 65 °C to reverse the crosslink, the DNA was purified with phenol/chloroform, precipitated with ethanol in the presence of 20 µg glycogen (Roche), resuspended in 10 mM Tris-HCl, pH 8.5 and then used for PCR or qPCR. Primers for PCR and real-time PCR are listed in Supplementary Tables 2 and 3. For PCR reactions, the products were run on a 2% agarose gel, stained with ethidium bromide and analyzed with ImageJ (NIH). Each PCR product of GAPDH from IP DNA was normalized to that from input DNA as internal control, giving rise to the ChIP efficiency. For real-time PCR, the threshold cycle (Ct) value of qPCR reactions for GAPDH of each IP DNA was normalized to that of input DNA as internal control,

giving rise to ChIP efficiency. Each PCR or qPCR product of the I-PpoI locus on chromosome 1 was first normalized to that from input DNA as internal control and then normalized to the corresponding ChIP efficiency. The y-axis in figures displaying ChIP results represents the relative occupancy normalized to the untreated control.

**Assays of DSB induction in ddI-PpoI cells.** Cells stably expressing pBabe-neo-ddI-PpoI were first treated with Shield-1 (1 µM) for 3 h and then with 4-OHT (1 µM) for 15 min. After washing twice in PBS, cells were collected and genomic DNA was isolated using the Gentra Puregene Cell kit (Qiagen) according to the man-ufacturer's protocol. qPCR was performed using primers (Supplementary Table 3) flanking the I-PpoI site on chromosome 1. The Ct values of qPCR from I-PpoI site was then normalized to the Ct values of qPCR from the GAPDH gene using the ΔΔCt method, giving rise to the percentage of the I-PpoI-induced DSB on chro-mosome 1 as described[18].

**Immunofluorescence.** Immunofluorescence (IF) was performed as described[26, 27]. All cell images were recorded on a Zeiss Axioplan 2 microscope with a Hamma-matsu C4742–95 camera and processed in Open Lab.

To quantify recruitment of BRCA1, RIF1 and SMARCAD1 to FokI-induced DSBs, fixed cells were co-immunostained with anti-BRCA1, anti-RIF1 or anti-SMARCAD1 antibody in conjunction with γH2AX. The γH2AX signal was used to mark the area of FokI-induced damage and the intensity of BRCA1, RIF1 or SMARCAD1 within the marked area was measured. To quantify the intensity of Myc-CSB at FokI-induced DSBs in RIF1-depleted cells, the mCherry signal was used to mark the area of damage and the intensity of Myc-CSB within that area was measured. The intensity of BRCA1, RIF1, SMARCAD1 or Myc-CSB at FokI-induced DSBs marked by γH2AX or mCherry was normalized respectively to their intensity of the same size area but away from the FokI-induced damage site in the same nucleus, giving rise to normalized signal intensity. All images for a given experiment were captured on the same day with the same exposure time. All analyses were carried out on unmodified images with ImageJ software (NIH). Data were represented as scatter plot graphs with the mean indicated. P values were derived using a two-tailed Mann–Whitney test.

**Recombinant CSB proteins and in vitro kinase assays.** Production of 6xHis-tagged wild-type and mutant CSB carrying amino acids from 2 to 322 was carried out essentially as described[59, 63] with minor modifications. Induction of CSB proteins was carried out overnight with 1 mM isopropylthiogalactoside at room temperature. The cell pellet was resuspended in Binding buffer [20 mM Tris-HCl pH 8.0, 500 mM NaCl, 10 mM imidazole, 1 mM PMSF] and lysed by sonication. Triton X-100 was then added to 0.1% and the lysate was shaken at 4 °C for 30 min. Following centrifugation, the supernatant was incubated with nickel resin (Qiagen) at 4 °C for 2 h. The beads were washed once in Binding buffer, three times in Wash buffer [20 mM Tris-HCl, pH 8.0, 500 mM NaCl, 50 mM imidazole, 10 mM β-mercaptoethanol and 1 mM PMSF] and then eluted three times with an elution buffer [20 mM Tris-HCl, pH 8.0, 500 mM NaCl, 880 mM imidazole and 10 mM β-mercaptoethanol]. The elutions were combined and dialysed against a dialysis buffer [20 mM HEPES pH7.9, 500 mM NaCl, 20% glycerol, 3 mM MgCl₂ and 1 mM DTT]. For cyclin A/CDK2 kinase assays, 2.5 µg of His-tagged wild-type and mutant CSB fragments was incubated with or without 50 ng of active recombinant cyclin A/CDK2 (14–488, Millipore) in the presence of ATP according to the manufacturer's protocol.

**Clonogenic survival assays.** Clonogenic survival assays were done as described[27]. For olaparib treatment, cells were seeded in triplicate at 300 cells per 6-cm plate, except for that 600 cells were seeded for 2 µM olaparib treatment. For UV treat-ment, a range of 750–1500 cells were seeded for siControl whereas a range of 750–6000 cells were seeded for siCSA. Twenty-four hours post seeding, cells were treated with olaparib or UV. Ten days later, colonies were fixed and stained at room temperature for 10 min with a solution containing 50% methanol, 7% acetic acid and 0.1% Coomassie blue. Colonies consisting of more than 32 cells were scored. The number of colonies for each drug or UV treatment were normalized to the number of clonies for untreated cells, giving rise to the percentage of cell survival.

**Statistical analysis.** A Student's two-tailed unpaired t test was used to derive all P values except for where specified.

**Data availability.** All the data used in this study are available within the article, Supplementary files, or available from the authors upon request.

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

## Acknowledgements

We would like to thank Titia de Lange for phoenix cells, Jan Hoeijmakers for the RAD51 antibody, Roger Greenberg for U2OS-265 cells, André Nussenzweig and Jeremy Daniel for the GFP-PTIP expression construct. Sebastien Landry is thanked for generating U2OS-265 53BP1 KO cells. This work is supported by funding from Canadian Institutes of Health Research to D.D. (FDN143343) and to X.-D.Z. (MOP-285822). N.L.B. is a holder of Ontario Graduate Scholarship.

## Author contributions

N.L.B.: Initiated the project and performed the majority of the experiments. J.R.W.: Carried out the sequence alignment and computer modeling of WHD, cloned all CSB mutants, immunoprecipitated Flag-CSB, produced recombinant CSB and assisted with IR experiments. S.M.N.: Conducted the mass spectrometric analysis of Flag-CSB. N.M.: Conducted the FACS analysis of DR-GFP assays. All authors contributed to the data analysis and interpretation. X.-D.Z.: Designed the project and wrote the paper with N.L.B., J.R.W., and with input from other authors.

## Additional information

**Competing interests:** The authors declare no competing financial interests.

