## [Peer Review File · Nature Communications]

Reviewers' Comments:

Reviewer #1:

Remarks to the Author:

Batenburg et al. ATM and CDK2 control chromatin remodeler CSB to establish feedback inhibition of RIF1 in DNA DSB repair pathway choice

This report describes interesting results that link CSB to RIF1 regulation of BRCA1 recruitment to DNA damage, thereby regulating choice between NHEJ and HR DSB repair pathways. In general the data support the conclusions, and both the writing and figure presentation are high quality. The demonstration that CSB promotes chromatin remodeling at DSBs in vivo is particularly novel and important. The detailed molecular analysis of regulatory components (phosphorylation sites, kinases, protein-protein interaction domains) comprise a comprehensive data set built on a solid experimental design. Several issues should be addressed as noted below.

The results in Fig. 2 demonstrate that the CSB - RIF1 interaction is important for CSB recruitment to DNA damage (DSBs induced by restriction enzyme Fok1 delivered to a Lac operator array via fusion to LacI). In Fig. 2c-e, the authors use expressed deletion mutant CSB and RIF1 fragments to map the necessary/sufficient domains in each protein that mediate their interaction. The data clearly demonstrate that the RIF1 CTDIII C-terminal domain is necessary and sufficient to interact with the C-terminal region of CSB. However, the data are confusing. Why doesn't full length RIF1 promote CSB recruitment to damage (Fig. 2c)? This is surprising because a small fragment of RIF1 (RIF1-C) promotes CSB recruitment to damage, and of course full length RIF1 includes the RIF1-C domain. Also, the y-axes in Figs. 2c-e vary widely among the three panels....why does CSB recruitment to damage appear stronger when the shorter fragments are expressed than full length proteins? For example, only 8% CSB positive cells appear with full length CSB and RIF1-C, but >20% are positive with CSB-C and RIF1-C fragments? These unusual outcomes are not discussed.

Regarding the RIF1 recruitment data, the authors present cell synchronization results and make the statement: "...peaked to about 38% at 6 h post release when the majority of cells were in S phase (Supplementary Fig. S4a)." The data in Fig S4a appear to support this statement but it would be strengthened if values for S phase fractions (along with G1 and G2/M fractions) were reported for each time point. Continuing on this point, the authors state that "...RIF1 accumulation at FokI-induced DSBs was at the highest level in cells arrested at the G1/S boundary (0 h post release) (Fig. 3a)..." It is not clear from the cell cycle profiles that cells released from DTB were at the G1/S boundary, as opposed to early or mid G1. Either data (or a reference) to support this contention should be provided, or the text modified.

p. 11: it would be helpful to indicate the significance of the cyclin A +/- status when this endpoint is first described (fig. 5e).

Fig. 5h: statistics for at least one olaparib concentration should be indicated to support the contention that CSB N30 is important for survival (stats also needed for Fig. S6a-c, and fig. 7i). BRCA1/2 status of these cells should also be indicated in the text to give the reader a frame of reference for the olaparib expt. Finally, it appears that the N30 mutant plot has both black and grey lines connecting the data points - one should be eliminated.

Fig. 6d: It appears that there is pS10 signal in the ATM deficient cells. It clearly increases when ATM is restored, but it is unclear whether the low pS10 signal in ATM deficient cells is due to residual ATM activity or phosphorylation by another kinase. It would be helpful to indicate the precise defect in the ATM deficient cells - knock out? Hypomorph?

p. 13: the CDK consensus sequence should be indicated in Fig. S7a, to compare with the CSB

S158 sequence.

Fig. 7b: The asterisks mark the non-specific bands - it would be useful to indicate the precise position of the pS158 band as there are at least two bands that are not indicated by asterisks.

p. 13: "Analysis of synchronized cell lysates revealed that phosphorylation of CSB on S158 was detectable as early as 3 h post release from a double thymidine block, continued to increase as cells progressed through S/G2/M but declined when cells returned to G1, 16 h post release (Fig. 7b)." Increasing the contrast of this western blot would help support this statement.

Fig. 8d: the positioning of BRCA1 in the final part of this model (one associated with a nucleosome, one on naked DNA) suggests functional significance, but it's not clear what that would be. Is there any reason why only some nucleosomes are evicted? The legend gives no clue as to these features of the model, pointing only to the text, and the text doesn't illuminate either. The model implies that BRCA1 on chromatin blocks RIF1/53BP1, but I don't recall data to support this idea. In any case, the model needs to accurately reflect what was known and was revealed in the present study, and the text or legend needs to be expanded to explain more clearly how the data link to various aspects of the model.

Typos and minor comments:

p. 3: ...whereas HR uses a sister chromatid as a template to repair broken DNA... This is a bit of an overstatement as HR can use any homologous sequence: homolog, repeated elements on the same or different chromosomes, in addition to (preferred) sister chromatids.

Fig. 4c,d: indicate CSB KO in figure (not just KO).

p. 10 (and elsewhere ref 18 is cited): "...in agreement with previous observations that histones are removed from chromatin surrounding DNA DSBs to accommodate HR-mediated repair - ref 18." The first demonstrations of histone removal near DSBs were *Nat. Struct. Mol. Biol.* 14, 1165-1172 (2007) in mammalian cells, and *Nature* 438, 479-483 (2005) in yeast.

p. 12: ...an S10A mutation....

p. 12: cells whereas overexpression of Myc-CSB-S10D... [typo, overexpression]

Fig. 7c: Cdk1 - should be CDK1 for consistent nomenclature

p. 13: ... CSB carrying an S158A mutation...

Fig. 7f: symbol key overlaps with data.

Reviewer #2:

Remarks to the Author:

This study by Batenburg et al reports on CSB's functions during DSB repair in the context of RIF1 regulation, chromatin remodeling, and regulation by kinases. Using a LacR-FokI-mediated DSB system, the authors first show that CSB is recruited at DSBs in a RIF1-dependent manner, and present IP results to demonstrate interaction between the two proteins. Second, they identify domains (CTDIII of RIF1 and CTD-WHD of CSB) that are required for CSB's recruitment to the DSB and the subsequent release of RIF1. Localization kinetics of RIF1 with and without CSB shows that CSB-KO leads to slower removal of RIF1 and reduced accumulation of BRCA1. Thirdly, the authors present data on CSB's role in chromatin remodeling in vivo, showing that an active form of CSB is needed for histone eviction. Finally, the authors demonstrate the significance of the N-terminus of

CSB as an auto-regulator and a subject of damage-induced phosphorylation (S10, S158), where phosphorylation is dependent on ATM and cyclin A-CDK2, respectively.

This paper covers many aspects of CSB biology in DNA DSB repair, and the manuscript contains a substantial amount of rigorous data. Nonetheless, there are several points to be addressed, as it is unclear how this work fits in with the rest of the 53BP1/RIF1/BRCA1 literature.

Specific comments:

1. The authors need to include a negative control (i.e. CSA) in key assays to show that this is a TCR-independent function of CSB.
2. In Figure 1G, the IP shows that the extent of RIF1 IP-ed by CSB is low. In contrast, in Figure 1F, RIF1 IP's CSB with a higher efficiency. This discrepancy should be addressed in the text.
3. Is the interaction between CSB and RIF1 direct? This is critical and should be relatively straightforward to test (using GST fusion proteins, for example) given that the authors have mapped the interaction to small domains of each protein.
4. In their previous EMBO paper, the authors found that transcription inhibitors block CSB recruitment to DSBs. How does this fit into the authors' model?
5. In Figure 3, levels of LacR-FokI induced DSBs during the time should be presented.
6. REV7/MAD2L2 has been shown to function downstream of RIF1, but its function is unclear. Does CSB affect REV7/MAD2L2 recruitment?
7. Similarly, PTIP is thought to function downstream of 53BP1, in parallel with RIF1. Does loss of CSB affect PTIP recruitment?
8. The presented data clearly show ATPase activity and the N-terminus of CSB are required for histone eviction and chromatin remodeling over 8 kb of DNA. However, this doesn't necessarily mean that CSB indeed can remodel chromatin at DSB. It is possible that CSB can function in recruiting other factors or change DNA status in an ATP hydrolysis dependent manner, which is crucial for histone eviction. BRCA1-mediated ubiquitination of H2A has been linked to chromatin remodeling at the DSB site in conjunction with SMARCAD1. Is CSB a parallel pathway, or does it function with these proteins?
9. Regarding CSB NT-autorepression, the cited papers (Lake et al 2010 & Wang et al 2014) show inhibitory effects of the CSB N-terminus, while (Cho et al 2013) showed that the NT is required for chromatin remodeling activity of CSB. The authors use all of them as references for auto-repression and explained their observation on CSB delta N30 and phosphorylation. But if CSB is autorepressed by the N-terminus, how is it that removal of the N30 residues and S/A mutations that decrease interaction with the ATPase domain can diminish CSB activity? And how does phosphorylation of CSB-NT, which increases binding with ATPase domain, turn on CSB's activity? Can the CSB model in Figure 8 reconcile with the autorepression idea?
10. In Figure 3F and Page 8 the authors write "CSB...establishes a negative feedback loop of 53BP1-RIF1-mediated NHEJ pathway choice." Apparently, CSB-KO decreases BRCA1 recruitment of at the DSB moderately in all time points but doesn't change the overall recruitment pattern. Data demonstrating any repair efficiency change between HR and NHEJ by CSB-KO would strengthen this claim.
11. In Figure 7B, it is not clear which band is CSB-p158.

Reviewer #3:

Remarks to the Author:

This manuscript describes a nice and in general well-evidenced study of the mechanism through which the chromatin remodeler CSB regulates DSB repair choice. It adds significant new information about the role of CSB in promoting HR-mediated repair of double-strand breaks, which will be of interest to the field. Some aspects of the Results presented however need further attention (see points 1-5).

While mostly well-written, there were a few places in the manuscript where the text or figures are unclear or confusing (points i-v below).

1. The experiments mapping CTD-III as the Rif1 domain mediating CSF interaction are not convincing. No explanation is given for the fact that RIF1-FL does not interact with CSB as would be expected. There is a concern that differences simply reflect varying expression or recruitment levels of the bait protein, especially with the very low % values of cells showing interaction in Fig 2c. For this reason, for the authors to conclude that Rif1 CTD-III is the critical interaction domain, it's important to map the domain in a co-IP experiment (similar to those shown in Fig. 1g/h, but testing the Rif1 subfragments).

2. More analysis and explanation of cell cycle dynamics is needed in the synchronisation experiments of Fig. 3. The authors state that by 16 hr the cells have returned to G1, but provide no evidence to support this. In Fig. 3e, RIF1 recruitment to the induced DSBs is elevated in the CSB knockout cells, but by 12 hr the proportion of cells with RIF1 foci is the same as in the WT. If CSB-RIF1 interaction is important for preventing RIF1/53BP1 association then in the absence of CSB how does the proportion of cells with RIF1 foci eventually gets back to WT levels?

3. It produces a misleading amplification of some effects to show some y-axes starting at zero and other at 0.5. All the y-axes should run from zero.

While there is a lot of scatter as is typical in such experiments, the ChIP data is convincing overall given the number of sites tested. Do the authors think that the relatively low cut frequency (Fig. S5d) contributes to the fairly low effect on histone eviction in ChIP? If so, that point could be made in the text.

4. The cyclical nature of the CSB-S158 phosphorylation in Fig. 7B is not convincing partly due to the dark background in the 10 hr time point (what is the band appearing high up close to the 250 kD marker in this sample?). A better gel and some quantitation are needed here.

5. In Fig 7g, the effect is tested of the S158 site proposed to be phosphorylated by CDK-cyclin A. Because it does not need cyclin A-CDK phosphorylation, the expectation is that the S158D phosphomimic mutant would suppress RIF1 focus formation regardless of the Cyclin A status of the cells. Why does it not?

i. In the second sentence of the Abstract, it's unclear whether the newly identified winged helix is in CSB, RIF1, or 53BP1.

ii. The term 'feedback inhibition' is confusing in the title and throughout the paper, because 'feedback' implies that the target (Rif1) somehow amplifies the signal from CSB, which is not what the authors are proposing. This leaves the reader searching, in the Abstract and through the text, for some additional aspect of the model involving such feedback. I suggest removing the term 'feedback' from the title and the rest of the paper.

iii. The x-axis annotation of Fig. 1d (and Fig. 3d) is confusing, in marking the absence of a protein

by a '+' sign.

iv. In Fig. 5e, 6g, and 7g the proportion of cells with RIF1 foci is shown in cells either positive or negative for Cyclin A. The initial and obvious interpretation is that Cyclin A here is being depleted as part of the experimental test. But in fact it is being used as a cell cycle marker, which needs to be explained.

v. In Fig. 8d it is unclear whether BRCA1 prefers to interact with nucleosomal or non-nucleosomal DNA. Some more background explanation of the effect of nucleosomes on BRCA1 interaction is needed.

The arrow in Fig. 8d indicating ATM phosphorylation points towards RIF1. Presumably the authors intend it to point at the purple (S10) phosphorylation site on CSB?

Point-by point response to the reviewers:

From Reviewer #1:

Point #1: “... However, the data are confusing. Why doesn't full length RIF1 promote CSB recruitment to damage (Fig. 2c)? This is surprising because a small fragment of RIF1 (RIF1-C) promotes CSB recruitment to damage, and of course full length RIF1 includes the RIF1-C domain. Also, the y-axes in Figs. 2c-e vary widely among the three panels....why does CSB recruitment to damage appear stronger when the shorter fragments are expressed than full length proteins? For example, only 8% CSB positive cells appear with full length CSB and RIF1-C, but >20% are positive with CSB-C and RIF1-C fragments? These unusual outcomes are not discussed.”

My response: We apologize for the confusion. We would like to clarify that full-length RIF1 does promote CSB recruitment to FokI-induced DSBs (Fig. 2h). The experiments in the original Fig. 2c-e address the recruitment of CSB to the lac operator by RIF1 fused to mCherry-LacR in the absence of induction of DSBs by FokI. In this experimental set-up, we did not observe robust interaction of full-length RIF1 and CSB at the lac operator array. In the revised manuscript, we have included the data showing that the expression of mCherry-LacR-RIF1-FL was much lower than that of mCherry-LacR-RIF1-N and mCherry-LacR-RIF1-C (Supplementary Fig. 1c), which likely contributed to the poor interaction observed between mCherry-LacR-RIF1-FL and Myc-CSB.

Point #2: “Regarding the RIF1 recruitment data, the authors present cell synchronization results and make the statement: “...peaked to about 38% at 6 h post release when the majority of cells were in S phase (Supplementary Fig. S4a).” The data in Fig S4a appear to support this statement but it would be strengthened if values for S phase fractions (along with G1 and G2/M fractions) were reported for each time point. Continuing on this point, the authors state that “...RIF1 accumulation at FokI-induced DSBs was at the highest level in cells arrested at the G1/S boundary (0 h post release) (Fig. 3a)...” It is not clear from the cell cycle profiles that cells released from DTB were at the G1/S boundary, as opposed to early or mid G1. Either data (or a reference) to support this contention should be provided, or the text modified.”

My response: We have included the values for G1, S and G2/M fractions for each time point as suggested in the revised manuscript. At 0 hr post release from a double thymidine block, the majority of cells were in G1 (not G1/S boundary). We apologize for this error. We have revised the text accordingly.

Point #3: “p. 11: it would be helpful to indicate the significance of the cyclin A +/- status when this endpoint is first described (fig. 5e).”

My response: We have previously reported that CSB suppresses IR-induced RIF1 foci formation specifically in S/G2 cells [Batenburg et al. (2015) *EMBO J*], which stain positive for cyclin A. We have included this information prior to describing Fig. 5e in the revised manuscript.

Point #4: “Fig. 5h: statistics for at least one olaparib concentration should be indicated to support the contention that CSB N30 is important for survival (stats also needed for Fig. S6a-c, and fig. 7i). BRCA1/2 status of these cells should also be indicated in the text to give the reader a frame of reference for the olaparib expt. Finally, it appears that the N30 mutant plot has both black and grey lines connecting the data points - one should be eliminated.”

My response: We apologize for the confusion of black and grey lines. We have revised the figure legend of original Fig. 5h (now Fig. 6h in the revised manuscript) to clarify that the dashed black line represents CSB- Δ N30 whereas the grey line represents the vector alone. In addition, the solid black line represents CSB. We have marked olaparib concentrations at which CSB- Δ N30 is statistically significantly different from wild type CSB but not from the vector alone in Fig. 6h. We have also revised original Supplementary Fig. 6a-c and Fig. 7i accordingly. Supplementary Fig. 6a-c are now Supplementary Fig. 8a-8c in the revised manuscript whereas Fig. 7i is now Fig. 8i in the revised manuscript.

Loss of CSB does not affect the expression of BRCA1 and BRCA2 (Batenburg et al. 2015; Batenburg and Zhu, unpublished data). We have cited this information in the revised manuscript.

Point #5: *“Fig. 6d: It appears that there is pS10 signal in the ATM deficient cells. It clearly increases when ATM is restored, but it is unclear whether the low pS10 signal in ATM deficient cells is due to residual ATM activity or phosphorylation by another kinase. It would be helpful to indicate the precise defect in the ATM deficient cells - knock out? Hypomorph?”*

My response: The ATM deficient GM16666A cells harbor a homozygous frameshift mutation at codon 762 of the ATM gene and no ATM protein is detected in these cells (Fig. 6d and [Ziv et al (1997) Oncogene]). Therefore the residual signal of CSB-pS10 in the ATM deficient cells might be due to the activity of another kinase. We have included this information in the revised manuscript.

Point #6: *“p. 13: the CDK consensus sequence should be indicated in Fig. S7a, to compare with the CSB S158 sequence.”*

My response: We have included the CDK consensus motif in the text as well as in Supplementary Fig. 9a in the revised manuscript.

Point #7: *“Fig. 7b: The asterisks mark the non-specific bands - it would be useful to indicate the precise position of the pS158 band as there are at least two bands that are not indicated by asterisks.”*

My response: We have repeated the experiment as requested by Reviewer #3 and included this new data as Fig. 8b in the revised manuscript. We have marked the position of CSB-pS158 band with an arrow.

Point #8: *“p. 13: “Analysis of synchronized cell lysates revealed that phosphorylation of CSB on S158 was detectable as early as 3 h post release from a double thymidine block, continued to increase as cells progressed through S/G2/M but declined when cells returned to G1, 16 h post release (Fig. 7b).” Increasing the contrast of this western blot would help support this statement.”*

My response: We have repeated the cell cycle experiment with anti-CSB-pS158 antibody (Fig. 8b) and quantified the CSB-pS158 signal from two independent experiments (Supplementary Fig. 9b and 9c). These data revealed that CSB-pS158 signal started to arise above the background level reproducibly at 6 h post release from a double thymidine block. We have revised the text accordingly.

Point #9: *“Fig. 8d: the positioning of BRCA1 in the final part of this model (one associated with a nucleosome, one on naked DNA) suggests functional significance, but it’s not clear what that would be. Is there any reason why only some nucleosomes are evicted? The legend gives no clue as to these*

features of the model, pointing only to the text, and the text is doesn't illuminate either. The model implies that BRCA1 on chromatin blocks RIF1/53BP1, but I don't recall data to support this idea. In any case, the model needs to accurately reflect what was known and was revealed in the present study, and the text or legend needs to be expanded to explain more clearly how the data link to various aspects of the model.”

My response: The work presented here does not address the interaction of BRCA1 with nucleosomal or non-nucleosomal DNA. The initial positioning of BRCA1 in the model was merely meant to depict that chromatin remodeling by CSB paves for the way for BRCA1-mediated HR. Clearly the drawing caused much confusion and we apologize for this confusion. We have removed the original positioning of BRCA1 in the revised model.

On average 45-50% of loss of histones is observed 2 h post induction of I-PpoI and therefore we included some nucleosomes in the original model. In the revised manuscript, we have described the model to accurately reflect what was reported in the present study.

Point #10: *“p. 3: ...whereas HR uses a sister chromatid as a template to repair broken DNA... This is a bit of an overstatement as HR can use any homologous sequence: homolog, repeated elements on the same or different chromosomes, in addition to (preferred) sister chromatids.”*

My response: We have revised the sentence to “whereas HR uses homologous sequences as a template to repair broken DNA” in the revised manuscript.

Point #11: *“Fig. 4c,d: indicate CSB KO in figure (not just KO).”*

My response: We have used “CSB KO” in Fig. 5c and 5d in the revised manuscript.

Point #12: *“p. 10 (and elsewhere ref 18 is cited): “...in agreement with previous observations that histones are removed from chromatin surrounding DNA DSBs to accommodate HR-mediated repair - ref 18.” The first demonstrations of histone removal near DSBs were Nat. Struct. Mol. Biol. 14, 1165-1172 (2007) in mammalian cells, and Nature 438, 479-483 (2005) in yeast.”*

My response: We apologize that we missed these references. We have included them in the revised manuscript.

Point #13, *“p. 12: ...an S10A mutation....”*

My response: We have revised the phrase.

Point #14: *“p. 12: cells whereas overexpression of Myc-CSB-S10D... [typo, overexpression]”*

My response: We have corrected it.

Point #15: *“Fig. 7c: Cdk1 - should be CDKi for consistent nomenclature”*

My response: We have corrected it.

Point #16: *“p. 13: ... CSB carrying an S158A mutation...”*

My response: We have revised the phrase.

Point #17: “Fig. 7f: symbol key overlaps with data.”

My response: We have fixed it.

From Reviewer #2:

Point #1: “*The authors need to include a negative control (i.e. CSA) in key assays to show that this is a TCR-independent function of CSB.*”

My response: The term “TCR” is commonly used to refer to transcription-coupled nucleotide excision repair, also known as TC-NER. TC-NER is responsible for repairing UV-induced DNA damage such as cyclobutane pyrimidine dimers (CPDs) and pyrimidine (6-4) pyrimidone photoproducts (6-4PPs). CSB interacts with CSA and they both play a role in TC-NER.

a) If this reviewer refers to TCR as TC-NER, we believe that it is inappropriate to use CSA as a negative control in our study. CSA is reported to localize to sites of DNA DSBs [Iyama and Wilson III (2016) *J Mol Biol*], indicating that CSA itself may play a role in DNA DSB repair. CSB is known to interact with CSA [reviewed in Aamann et al. (2013) *Mech Age Dev*]. Whether this interaction is important for DNA DSB repair requires future investigation and is beyond the scope of the submitted manuscript. Aside from CSA, several other NER proteins have also been implicated in homologous recombinational repair, including XPG [Trego et al (2016) *Mol Cell*] and XPF [Ahmad et al (2008) *Mol Cell Biol*]. Therefore in general, we believe that it is inappropriate to use NER factors as a negative control in our study.

b) In the submitted manuscript, we examined the role of CSB in DNA DSB repair pathway choice at FokI- and IPpoI-induced DNA double strand breaks (DSBs). The use of restriction enzymes FokI and IPpoI in our study eliminates any ambiguity about the nature of DNA damage being DNA DSBs. Therefore the function of CSB described in the submitted manuscript is specifically about its role at DNA DSBs, regardless of its function in TC-NER.

c) Recent studies suggest that there exists a sub-path of DSB repair called transcription-coupled DNA DSB repair [see review by Marnerf et al (2017) *J Mol Biol*]. We have previously reported that CSB recruitment to sites of DNA DSBs is transcription-dependent [Batenburg et al (2015) *EMBO J*]. Whether CSB is involved in transcription-coupled DNA DSB repair would require future investigation and is beyond the scope of our current manuscript.

Point #2: “*In Figure 1G, the IP shows that the extent of RIF1 IP-ed by CSB is low. In contrast, in Figure 1F, RIF1 IP’s CSB with a higher efficiency. This discrepancy should be addressed in the text.*”

My response: The discrepancy between the amount of CSB brought down by anti-RIF1 antibody and the amount of RIF1 brought down by anti-CSB antibody may imply that CSB might not interact with RIF1 in a 1:1 stoichiometry, however we cannot rule out the possibility that this discrepancy may be due to a difference in IP efficiency. We have included this discussion in the revised manuscript.

Point #3: *“Is the interaction between CSB and RIF1 direct? This is critical and should be relatively straightforward to test (using GST fusion proteins, for example) given that the authors have mapped the interaction to small domains of each protein.”*

My response: We have observed that the winged helix domain (WHD) of CSB is necessary but not sufficient for CSB interaction with RIF1. While the C-terminal domain of CSB (CSB-C) spanning amino acids 972-1493 is sufficient to interact with RIF1, we have not been able to produce recombinant CSB-C protein in bacteria, which prohibits us from examining the direct interaction of CSB-C with RIF1-CTD.

As suggested by Reviewer #3, we have performed coimmunoprecipitation experiments using Myc-CSB-C and various mCherry-LacR-RIF1-CTD alleles and included the new data in the revised manuscript (Fig. 2f). We have shown that the CTD domain alone of RIF1 was sufficient to interact with Myc-CSB-C via coimmunoprecipitation. While deletion of the CTDI subdomain had a moderate effect on the interaction of mCherry-LacR-RIF1-CTD with Myc-CSB-C, deletion of the CTDIII subdomain severely impaired its interaction with Myc-CSB-C, supporting the notion that CTDIII is necessary for RIF1 interaction with CSB.

In the original manuscript we have reported a robust interaction between mCherry-LacR-RIF1-CTDIII and Myc-CSB-C at the lac operator array (see Fig. 2f in the revised manuscript). However mCherry-LacR-RIF1-CTDIII was not substantially enriched in Myc-CSB-C immuno-complex (Fig. 2f). These findings suggest that the CTDIII subdomain alone may not be sufficient to mediate RIF1 interaction with CSB. The observed discrepancy may be due in part to the difference in the experimental conditions. Nevertheless, our data suggest that the CTD domain alone is necessary and sufficient to mediate RIF1 and CSB. We have revised the manuscript accordingly.

Point #4: *“In their previous EMBO paper, the authors found that transcription inhibitors block CSB recruitment to DSBs. How does this fit into the authors’ model?”*

My response: It has been reported that RNA mediates 53BP1 and RIF1 recruitment to sites of DSB [Li et al. (2017) *DNA Repair*; Pryde et al (2005) *J. Cell Sci*]. It is possible that CSB recruitment by RIF1 to sites of DSBs might be regulated by transcription, which would require future investigation. We have included this discussion in the revised manuscript.

Point #5: *“In Figure 3, levels of LacR-FokI induced DSBs during the time should be presented.”*

My response: In the revised manuscript, we have included new data showing γ H2AX localization at FokI-induced DSBs throughout the cell cycle (Fig. 4b). The levels of FokI-induced DSBs did not alter significantly throughout the cell cycle. In addition, knockout of 53BP1 or CSB had little effect on the level of FokI-induced DSBs (Fig. 4d and 4g).

Point #6: *“REV7/MAD2L2 has been shown to function downstream of RIF1, but its function is unclear. Does CSB affect REV7/MAD2L2 recruitment?”*

My response: We have performed the suggested experiment and found that knockout of CSB resulted in an increase in MAD2L2 accumulation at FokI-induced DSBs. We have included this new data in Fig. 3d in the revised manuscript.

Point #7: *“Similarly, PTIP is thought to function downstream of 53BP1, in parallel with RIF1. Does*

loss of CSB affect PTIP recruitment?”

My response: We have performed the suggested experiment and found that knockout of CSB does not significantly affect PTIP recruitment to FokI-induced DSBs. We have included this new data in Fig. 3e. Our data suggest that CSB restricts the RIF1-MAD2L2 pathway but not the parallel PTIP pathway.

Point #8: “*The presented data clearly show ATPase activity and the N-terminus of CSB are required for histone eviction and chromatin remodeling over 8 kb of DNA. However, this doesn’t necessarily mean that CSB indeed can remodel chromatin at DSB. It is possible that CSB can function in recruiting other factors or change DNA status in an ATP hydrolysis dependent manner, which is crucial for histone eviction. BRCA1-mediated ubiquitination of H2A has been linked to chromatin remodeling at the DSB site in conjunction with SMARCAD1. Is CSB a parallel pathway, or does it function with these proteins?”*

My response: Technically it is not feasible to distinguish if CSB removes histones directly on its own or through changing DNA status in an ATP hydrolysis dependent manner *in vivo*. It is well established that CSB remodels chromatin *in vitro* [Lake et al. (2010) *Mol Cell*; Cho et al (2013) *PLoS Genetics*; Citterio et al (2000) *Mol Cell Biol*]. In this study, we directly measured histone occupancy in CSB null cells and found that CSB null cells fail to remove histones from chromatin surrounding a single DSB (Fig. 5). We have shown that loss of CSB impairs BRCA1 accumulation at FokI-induced DSBs (Fig. 3f). As this reviewer pointed out, BRCA1 mediates ubiquitylation of H2A that is recognized by chromatin remodeler SMARCAD1. Therefore formally it was possible that the loss of histone displacement in CSB null cells might have resulted from impaired recruitment of SMARCAD1. However we did not detect any significant change in SMARCAD1 recruitment to FokI-induced DSBs in CSB null cells (Supplementary Fig. 7a and 7b in the revised manuscript). Taken together, these findings argue that CSB is also a chromatin remodeler *in vivo*. Furthermore, our finding suggests that CSB may act independently of chromatin remodeler SMARCAD1 in promoting HR-mediated DSB repair.

Point #9: “*Regarding CSB NT-autorepression, the cited papers (Lake et al 2010 & Wang et al 2014) show inhibitory effects of the CSB N-terminus, while (Cho et al 2013) showed that the NT is required for chromatin remodeling activity of CSB. The authors use all of them as references for auto-repression and explained their observation on CSB delta N30 and phosphorylation. But if CSB is autorepressed by the N-terminus, how is it that removal of the N30 residues and S/A mutations that decrease interaction with the ATPase domain can diminish CSB activity? And how does phosphorylation of CSB-NT, which increases binding with ATPase domain, turn on CSB’s activity? Can the CSB model in Figure 8 reconcile with the autorepression idea?”*

My response: We have extensively revised the discussion on the CSB model in the revised manuscript. We envision a model in which in the absence of DSBs, the N-terminal region of CSB is docked on its ATPase domain in such a manner that its ATPase activity is restricted. Upon induction of DSBs, ATM- and CDK2-dependent CSB phosphorylations on S10 and S158 promote CSB conformational change so that its N-terminal region of CSB is now docked at a different location on its ATPase domain (Fig. 9d). This new docking frees its ATPase activity needed for its chromatin remodeling activity. In the absence of these two phosphorylation events, CSB-ΔN30, CSB-S10A and CSB-S158A mutants would not be able to undergo protein conformational change needed for the new docking of its N-terminal region. Our finding that CSB phosphorylations on S10A and S158 stimulate CSB-N interaction with CSB-ATPase suggests that these two phosphorylation events might create an interface that is more favorable for these

two domains to interact. We propose that these two phosphorylation events act together as molecular signals to trigger the release of the autoinhibition of its N-terminal region on its ATPase domain in S/G2 cells (Fig. 9d). Subsequently the chromatin remodeling activity of CSB at DSBs evicts histones and disassembles nucleosomes, which limits RIF1 accumulation and paves the way for BRCA1-mediated HR activity (Fig. 9d).

Point #10: “In Figure 3F and Page 8 the authors write “CSB...establishes a negative feedback loop of 53BP1-RIF1-mediated NHEJ pathway choice.” Apparently, CSB-KO decreases BRCA1 recruitment of at the DSB moderately in all time points but doesn't changes the overall recruitment pattern. Data demonstrating any repair efficiency change between HR and NHEJ by CSB-KO would strengthen this claim.”

My response: We have performed additional experiments and shown that loss of CSB impairs HR- but promotes NHEJ-mediated repair of I-SceI-induced DSBs. We have included this new data in Fig. 3g and 3h.

Point #11: “In Figure 7B, it is not clear which band is CSB-p158.”

My response: We have repeated the experiment in Figure 7b and included the new data as Fig. 8b in the revised manuscript. We have marked the position of the CSB-p158 band with an arrow in the revised manuscript.

From Reviewer #3:

Point #1: “The experiments mapping CTD-III as the Rif1 domain mediating CSF interaction are not convincing. No explanation is given for the fact that RIF1-FL does not interact with CSB as would be expected. There is a concern that differences simply reflect varying expression or recruitment levels of the bait protein, especially with the very low % values of cells showing interaction in Fig 2c. For this reason, for the authors to conclude that Rif1 CTD-III is the critical interaction domain, it's important to map the domain in a co-IP experiment (similar to those shown in Fig. 1g/h, but testing the Rif1 subfragments).”

My response: We have included the new data showing that the expression of mCherry-LacR-RIF1-FL was much lower than that of mCherry-LacR-RIF1-N and mCherry-LacR-RIF1-C (Supplementary Fig. 1c), which likely contributed to the poor interaction observed between mCherry-LacR-RIF1-FL and Myc-CSB.

We have also performed suggested coimmunoprecipitation experiment using Myc-CSB-C and various mCherry-LacR-RIF1-CTD alleles and included the new data in the revised manuscript (Fig. 2f). We have shown that the CTD domain alone of RIF1 was sufficient to interact with Myc-CSB-C. While deletion of the CTDI subdomain had a moderate effect on the interaction of mCherry-LacR-RIF1-CTD with Myc-CSB-C, deletion of the CTDIII subdomain severely impaired its interaction with Myc-CSB-C, supporting the notion that CTDIII is necessary for RIF1 interaction with CSB.

In the original manuscript we have reported a robust interaction between mCherry-LacR-RIF1-CTDIII and Myc-CSB-C at the lac operator array (see Fig. 2f in the revised manuscript). However mCherry-LacR-RIF1-CTDIII was not substantially enriched in Myc-CSB-C immuno-complex (Fig. 2f). These findings suggest that the CTDIII subdomain alone may not be sufficient to mediate RIF1

interaction with CSB. The observed discrepancy may be due in part to the difference in the experimental conditions. Nevertheless, our data suggest that the CTD domain alone is necessary and sufficient to mediate RIF1 and CSB. We have revised manuscript accordingly.

Point #2: *“More analysis and explanation of cell cycle dynamics is needed in the synchronisation experiments of Fig. 3. The authors state that by 16 hr the cells have returned to G1, but provide no evidence to support this. In Fig. 3e, RIF1 recruitment to the induced DSBs is elevated in the CSB knockout cells, but by 12 hr the proportion of cells with RIF1 foci is the same as in the WT. If CSB-RIF1 interaction is important for preventing RIF1/53BP1 association then in the absence of CSB how does the proportion of cells with RIF1 foci eventually gets back to WT levels?”*

My response: In the revised manuscript, we have included the values for G1, S and G2/M fractions in the cell cycle profile obtained for cells collected at 0h, 2h, 4h, 6h, 8h, 10h, 12h and 16h (see Supplementary Fig. 5).

In this manuscript and in our previous publication [Batenberg et al. (2015) *EMBO J*], we have demonstrated that CSB inhibits RIF1 accumulation specifically in S/G2. At 12 h post release from a double thymidine block, cells were exiting G2/M and enriched in G1. That the proportion of cells with RIF1 foci is the same as in the WT at 12 h post release is in agreement with the notion that CSB inhibits RIF1 accumulation specifically in S/G2. We have included this information in the revised manuscript.

Point #3: *“It produces a misleading amplification of some effects to show some y-axes starting at zero and other at 0.5. All the y-axes should run from zero. While there is a lot of scatter as is typical in such experiments, the ChIP data is convincing overall given the number of sites tested. Do the authors think that the relatively low cut frequency (Fig. S5d) contributes to the fairly low effect on histone eviction in ChIP? If so, that point could be made in the text.”*

My response: We have revised all y-axes and they all now start at zero in the revised manuscript. We observed that histone eviction in wild type cells peaked 2 h post induction of I-PpoI, with on average 45-50% of loss of histones on chromatin surrounding the I-PpoI cleavage site (Fig. 5a and 5b). This effect is very similar to that previously reported [Goldstein et al. (2013) *PNAS*; Li and Tyler (2016) *eLIFE*]. Therefore we don't believe that the effect of I-PpoI induction on histone eviction is particularly low compared to what has been reported. Two-hour post induction, the frequency of I-PpoI-induced cleavage on chromosome 1 in hTERT-RPE cells was on average 21%, close to a previously-reported frequency of ~30% at this locus in MCF7 cells [Goldstein et al. (2013) *PNAS*]. Perhaps, the frequency of I-PpoI cleavage on the *DABI* locus varies depending upon the cell type. We have included this discussion in the revised manuscript.

Point #4: *“The cyclical nature of the CSB-S158 phosphorylation in Fig. 7B is not convincing partly due to the dark background in the 10 hr time point (what is the band appearing high up close to the 250 kD marker in this sample?). A better gel and some quantitation are needed here.”*

My response: We have repeated the experiment and replaced the original Fig. 7B with the new Fig. 8b in the revised manuscript. We have also included the quantification data for two independent experiments in Supplementary Fig. 9b and 9c in the revised manuscript. In addition, we have included new western analysis with CSB-pS158 antibody in wild type and CSB knockout cells that were either asynchronous or 10 hr post release from a double thymidine block (Supplementary Fig. 9d), demonstrating that the band close to 250 kD marker is present in CSB knockout cells and thus non-

specific.

Point #5: *“In Fig 7g, the effect is tested of the S158 site proposed to be phosphorylated by CDK-cyclin A. Because it does not need cyclin A-CDK phosphorylation, the expectation is that the S158D phosphomimic mutant would suppress RIF1 focus formation regardless of the Cyclin A status of the cells. Why does it not?”*

My response: In this study, we have shown that CSB is recruited to DSBs in S/G2 cells (Fig. 4a in the revised manuscript) and this recruitment is mediated via its C-terminus by RIF1. CSB phosphorylation on its N-terminal S158 controls its chromatin remodeling activity but does not control its recruitment to DSBs. Both CSB-S158A and CSB-158D are recruited to DSBs indistinguishably from wild type CSB (Supplementary 9f). Thus the CSB-S158D mutant suppresses RIF1 focus formation in S/G2 (cyclin A-positive) cells but not in G1 (cyclin A-negative) cells since it is not recruited to DSBs outside of S/G2.

Point #i: *“In the second sentence of the Abstract, it’s unclear whether the newly identified winged helix is in CSB, RIF1, or 53P1.”*

My response: We have revised the wording in the abstract.

Point #ii: *“The term ‘feedback inhibition’ is confusing in the title and throughout the paper, because ‘feedback’ implies that the target (Rif1) somehow amplifies the signal from CSB, which is not what the authors are proposing. This leaves the reader searching, in the Abstract and through the text, for some additional aspect of the model involving such feedback. I suggest removing the term ‘feedback’ from the title and the rest of the paper.”*

My response: We have adopted the recommendation and revised the title and the text accordingly.

Point #iii: *“The x-axis annotation of Fig. 1d (and Fig. 3d) is confusing, in marking the absence of a protein by a ‘+’ sign.”*

My response: We have revised the annotation in Fig. 1d and 4d (original 3d).

Point #iv: *“In Fig. 5e, 6g, and 7g the proportion of cells with RIF1 foci is shown in cells either positive or negative for Cyclin A. The initial and obvious interpretation is that Cyclin A here is being depleted as part of the experimental test. But in fact it is being used as a cell cycle marker, which needs to be explained.”*

My response: We have clarified the use of Cyclin A in the revised manuscript.

Point #v: *“In Fig. 8d it is unclear whether BRCA1 prefers to interact with nucleosomal or non-nucleosomal DNA. Some more background explanation of the effect of nucleosomes on BRCA1 interaction is needed. The arrow in Fig. 8d indicating ATM phosphorylation points towards RIF1. Presumably the authors intend it to point at the purple (S10) phosphorylation site on CSB?”*

My response: The work presented here does not address the interaction of BRCA1 with nucleosomal or non-nucleosomal DNA. The initial positioning of BRCA1 in the model was merely meant to depict that chromatin remodeling by CSB paves the way for BRCA1-mediated HR. Clearly the drawing caused

much confusion and we apologize for this confusion. Our data suggest that chromatin remodeling by CSB inhibits RIF1 and promotes BRCA1-mediated HR. We have revised the model and the text accordingly.

Reviewers' Comments:

Reviewer #1:

Remarks to the Author:

Batenburg et al. ATM and CDK2 control chromatin remodeler CSB to establish feedback inhibition of RIF1 in DNA DSB repair pathway choice

The authors have done an excellent job revising the manuscript and adding new data in response to the prior critiques. My only significant question regards the model in Fig. 9D. In the discussion the authors make the statement: "Upon induction of DSBs, ATM- and CDK2-dependent CSB phosphorylations on S10 and S158 promote CSB conformational change so that its N-terminal region is now docked at a different location on its ATPase domain (Fig. 9d). While I agree that it's reasonable to suggest these phosphorylation events alter CSB conformation, I don't believe there is any evidence suggesting the second part of this model, that is, the shift causes a new docking site for the N terminal region. In fact, the term "dock" is not mentioned in the text until this section of the Discussion. It is suggested that the text be revised to a more generalized presentation of the model, omitting specifics about docking sites (or at least pointing out that such specifics are speculative).

A few minor points:

p. 3: clarify: ...the chromatin structure needs to be modified to facilitate efficient access of repair factors to DSBs...

p. 3: clarify: ...but the physiological mechanism that permits release of its ATPase activity... By release, do the authors mean "activate"?

p 9: rephrase: ...This value increased sharply to about 30%

p. 11: Perhaps the cleavage frequency by I-PpoI may vary depending upon the cell type. Perhaps...may.... are redundant

p. 11: rephrase: ...or Myc-CSB carrying a W851R ATPase-dead mutation..

Reviewer #2:

Remarks to the Author:

In the revised manuscript, the authors have addressed many of the concerns raised, but critical experiments are still missing. The authors describe a novel role for CSB, but seem oddly reluctant to test other known TCR proteins such as CSA, XPF, XPG, etc. The argument that these need not be tested because others have reported connections between them and DSB repair pathways defies logic. To me, these hints in the literature make testing these proteins all the more critical and potentially interesting. These experiments will be important for establishing whether this is a novel, independent role of CSB or involves a broader pathway that requires the other TC-NER proteins, and will be necessary for publication in a top journal like Nature Communications. The authors also have been unable to test whether the CSB-RIF1 interaction is direct.

Point-by point response to the reviewers:

From Reviewer #1:

Point #1: “My only significant question regards the model in Fig. 9D. In the discussion the authors make the statement: “Upon induction of DSBs, ATM- and CDK2-dependent CSB phosphorylations on S10 and S158 promote CSB conformational change so that its N-terminal region is now docked at a different location on its ATPase domain (Fig. 9d). While I agree that it’s reasonable to suggest these phosphorylation events alter CSB conformation, I don’t believe there is any evidence suggesting the second part of this model, that is, the shift causes a new docking site for the N terminal region. In fact, the term “dock” is not mentioned in the text until this section of the Discussion. It is suggested that the text be revised to a more generalized presentation of the model, omitting specifics about docking sites (or at least pointing out that such specifics are speculative).”

My response: We have adopted this Reviewer’s suggestion and removed the term “dock”. We have revised this section to “We envision a model in which in the absence of DSBs, the CSB N-terminal region interacts with its ATPase domain in such a manner that its ATPase activity is restricted. Upon induction of DSBs, ATM- and CDK2-dependent CSB phosphorylation of S10 and S158 promote CSB conformational change, which releases the inhibitory effect of its N-terminal region on its ATPase activity (Fig. 9d). In the absence of these two phosphorylation events, CSB-ΔN30, CSB-S10A and CSB-S158A mutants would not be able to undergo protein conformational change needed for stimulation of its ATPase activity.”

Point #2: “p. 3: clarify: ...the chromatin structure needs to be modified to facilitate efficient access of repair factors to DSBs...”

My response: We have included the clarification “of repair factors” in the revised manuscript.

Point #3: “p. 3: clarify: ...but the physiological mechanism that permits release of its ATPase activity... By release, do the authors mean “activate”?”

My response: We have revised the phrase to “physiological mechanism that promotes its ATPase activity.”

Point #4: “p 9: rephrase: ...This value increased sharply to about 30%”

My response: We have made the correction from “number” to “value” as suggested.

Point #5: “p. 11: Perhaps the cleavage frequency by I-PpoI may vary depending upon the cell type. Perhaps...may.... are redundant”

My response: We have removed “may” in the revised manuscript.

Point #6: “p. 11: rephrase: ...or Myc-CSB carrying a W851R ATPase-dead mutation.”

My response: We have adopted this phrase in the revised manuscript.

From Reviewer #2:

Point #1: *“In the revised manuscript, the authors have addressed many of the concerns raised, but critical experiments are still missing. The authors describe a novel role for CSB, but seem oddly reluctant to test other known TCR proteins such as CSA, XPF, XPG, etc. The argument that these need not be tested because others have reported connections between them and DSB repair pathways defies logic. To me, these hints in the literature make testing these proteins all the more critical and potentially interesting. These experiments will be important for establishing whether this is a novel, independent role of CSB or involves a broader pathway that requires the other TC-NER proteins, and will be necessary for publication in a top journal like Nature Communications.”*

My response: We have come to appreciate this Reviewer’s suggestion to include CSA, a well-known TC-NER protein, as a control in our work. We have included these new data (Fig. 1g and Supplementary 4c-4f) in the revised manuscript.

We have shown that CSB is recruited to FokI-induced DSBs and specifically inhibits the RIF1-MAD2L2 pathway in the choice of DSB repair pathways. Using CSA as a control, we did not observe any detectable accumulation of endogenous CSA at FokI-induced DSBs (unpublished data, N.L. Batenburg and X.D. Zhu in the revised manuscript). In addition, depletion of CSA did not affect CSB recruitment to FokI-induced DSBs (Fig. 1g). Furthermore, while depletion of CSA sensitized cells to UV damage, it had little impact on the accumulation of RIF1 and MAD2L2 at FokI-induced DSBs (Supplementary Fig. 4c-4f in the revised manuscript). Taken together, these results suggest that CSA is not involved in inhibition of the RIF1-MAD2L2 pathway and that the function of CSB in DNA DSB repair pathway choice is unlikely to be a general feature of TCR proteins.

Point #2: *“The authors also have been unable to test whether the CSB-RIF1 interaction is direct.”*

My response: Although we have not been able to express recombinant C-terminal domain of CSB in bacteria cells that would allow us to examine the direct interaction of CSB-C with RIF1-CTD *in vitro*, we did include new data from additional coimmunoprecipitation experiments in the first round of the revision (Fig 2f). Therefore we have done extensively characterization of CSB interaction with RIF1 *in vivo*. Through analysis of immunofluorescence and coimmunoprecipitation, we have demonstrated that CSB interacts with RIF1 in human cells and that this interaction is mediated through the CTD of RIF1 and the winged helix domain of CSB. Furthermore, we have shown that this interaction mediates CSB recruitment to sites of DSBs, which in turn inhibits the RIF1-MAD2L2 pathway but promotes HR.

Reviewers' Comments:

Reviewer #2:

Remarks to the Author:

Authors have been responsive. The revised paper is nice and ready to go to press, I think.

Point-by point response to the reviewers:

From Reviewer #2:

Point #1: “Authors have been responsive. The revised paper is nice and ready to go to press, I think.”

My response: We agree that the revised paper is ready to go to press.